# Recurrent pregnancy loss is associated with a pro-senescent decidual response during the peri-implantation window

Emma S. Lucas[1,2,5], Pavle Vrljicak[1,2,5], Joanne Muter[1,2], Maria M. Diniz-da-Costa[1,2], Paul J. Brighton[2], Chow-Seng Kong[2], Julia Lipecki[3], Katherine J. Fishwick[2], Joshua Odendaal[2], Lauren J. Ewington ⓘ [2], Siobhan Quenby[1,2], Sascha Ott[1,4] & Jan J. Brosens ⓘ [1,2]*

During the implantation window, the endometrium becomes poised to transition to a pregnant state, a process driven by differentiation of stromal cells into decidual cells (DC). Perturbations in this process, termed decidualization, leads to breakdown of the feto-maternal interface and miscarriage, but the underlying mechanisms are poorly understood. Here, we reconstructed the decidual pathway at single-cell level in vitro and demonstrate that stromal cells first mount an acute stress response before emerging as DC or senescent DC (snDC). In the absence of immune cell-mediated clearance of snDC, secondary senescence transforms DC into progesterone-resistant cells that abundantly express extracellular matrix remodelling factors. Additional single-cell analysis of midluteal endometrium identified *DIO2* and *SCARA5* as marker genes of a diverging decidual response in vivo. Finally, we report a conspicuous link between a pro-senescent decidual response in peri-implantation endometrium and recurrent pregnancy loss, suggesting that pre-pregnancy screening and intervention may reduce the burden of miscarriage.

[1] Tommy's National Centre for Miscarriage Research, University Hospitals Coventry & Warwickshire, Coventry CV2 2DX, UK. [2] Division of Biomedical Sciences, Clinical Sciences Research Laboratories, Warwick Medical School, University of Warwick, Coventry CV2 2DX, UK. [3] School of Life Sciences, Gibbet Hill Campus, University of Warwick, Coventry CV4 7AL, UK. [4] Department of Computer Science, University of Warwick, Coventry CV4 7AL, UK. [5] These authors contributed equally: Emma S. Lucas, Pavle Vrljicak. *email: J.J.Brosens@warwick.ac.uk

Approximately 15% of clinical pregnancies result in miscarriage[1], most often during the first trimester. Foetal chromosomal abnormalities account for 50–60% of sporadic miscarriages[2], although the incidence is lower in recurrent pregnancy loss (RPL)[3,4], defined as two or more losses[5,6]. Further, with each additional miscarriage, the frequency of euploid loss increases whereas the likelihood of a successful pregnancy decreases[7], indicating that uterine factors drive higher-order miscarriages. Few interventions improve live-birth rates in RPL[5], reflecting that in most cases the underlying mechanisms are incompletely understood.

Following the postovulatory rise in circulating progesterone levels, the endometrium becomes transiently receptive to embryo implantation during the midluteal phase of the cycle. This implantation window heralds the start of intense tissue remodelling[8], driven by differentiation of endometrial stromal cells (EnSC) into decidual cells (DC) and accumulation of uterine natural killer (uNK) cells[9]. Upon embryo implantation, DC rapidly encapsulate the conceptus[10], engage in embryo biosensing[11], and then form a decidual matrix that controls trophoblast invasion[12]. At a molecular level, decidual transformation of EnSC encompasses genome-wide remodelling of the chromatin landscape[13], reprogramming of multiple signalling pathways[14–16], and activation of decidual gene networks[17,18]. This multistep differentiation process starts with an evolutionarily conserved acute cellular stress response[19], marked by a burst of reactive oxygen species (ROS) and release of proinflammatory cytokines[9,20,21]. After a lag period of several days, EnSC lose their fibroblastic appearance and emerge as secretory DC with abundant cytoplasm and prominent endoplasmic reticulum[8]. A hallmark of DC is their resistance to multiple stress signals. Several mechanisms underpin decidual stress resistance, including silencing of the c-Jun N-terminal kinase (JNK) pathway and upregulation of various stress defence proteins and ROS scavengers[8,15,22]. In addition, DC highly express 11β-hydroxysteroid dehydrogenase type 1[23], which converts inactive cortisone into cortisol, a potent anti-inflammatory glucocorticoid. Thus, compared with EnSC, DC are exquisitely adapted to withstand the hyperinflammation and stress associated with deep haemochorial placentation.

Recently, we demonstrated that decidualization also results in the emergence of senescent decidual cells (snDC), both in vitro and in vivo[9,24]. In the stroma, the abundance of cells expressing p16[INK4], a tumour suppressor and canonical senescence marker, peaks transiently during the midluteal phase before rising again prior to menstruation[9]. Cellular senescence is defined by a state of permanent cell-cycle arrest and prominent secretion of various bioactive molecules, including ROS, extracellular matrix (ECM) remodelling proteins, proinflammatory cytokines, chemokines and growth factors, referred to as senescence-associated secretory phenotype (SASP)[25–27]. Different types of senescent cells underpin pathological and physiological processes. Chronic senescent cells accumulate progressively in response to various stressors and cause gradual loss of organ function during ageing mediated by the deleterious effects of the SASP on tissue homoeostasis[25,27]. By contrast, acute senescent cells are linked to biological processes that involve programmed tissue remodelling, including embryogenesis and wound healing[27–29]. They are induced in response to specific signals, produce a transient SASP with defined paracrine functions, and are promptly cleared by immune cells[25]. Recently we demonstrated that snDC exhibit hallmarks of acute senescent cells[9]. First, DC and snDC both emerge in response to FOXO1 activation, a pivotal decidual transcription factor downstream of the protein kinase A (PKA) and progesterone signalling pathways. Second, their associated SASP critically amplifies the initial decidual inflammatory response, which not only drives differentiation of EnSC but is also linked to induction of key receptivity genes[21]. Further, we have provided evidence that DC recruit and activate uNK cells, which in turn may eliminate snDC through perforin- and granzyme-containing granule exocytosis[9,24].

Although snDC constitute a relatively minor and variable stromal population in midluteal endometrium, they have the potential to impact profoundly on the unfolding decidual response in a manner that either promotes or precludes pregnancy progression. For example, a transient SASP associated with acute senescent cells has been shown to promote tissue plasticity by expanding resident progenitor populations[9,30]. By contrast, senescent cells that persist (i.e. chronic senescent cells) can induce senescence in neighbouring cells though juxtacrine signalling (termed 'secondary' or 'bystander' senescence), leading to spatiotemporal propagation of the phenotype and loss of tissue function[25,31,32].

Recently we reported loss of clonal mesenchymal stem-like cells (MSC) in midluteal endometrium of RPL patients but how this is linked to subsequent breakdown of the decidual–placental interface in pregnancy is unclear[33]. We hypothesized that MSC deficiency may drive a pro-senescent decidual response during the peri-implantation window, culminating in chronic inflammation in early pregnancy, proteolysis of the decidual–placental interface, and miscarriage. To test this conjecture, we first performed single-cell transcriptomic analysis on decidualizing primary EnSC cultures, mapped the emergence of DC and snDC, and identified co-regulatory gene networks underpinning the multistep decidual pathway. We then extended the single-cell analysis to peri-implantation endometrial biopsies and screened for stroma-specific marker genes that signal divergence of the decidual response towards a pro-senescent decidual state. The expression of two marker genes, DIO2 and SCARA5, and the abundance of uNK cells were then analysed in peri-implantation endometrial biopsies from 89 RPL patients and 90 control subjects.

## Results

**Single-cell analysis of the decidual pathway in vitro.** To identify putative marker genes of DC and snDC, we first reconstructed the decidual pathway in vitro using single-cell transcriptomics. As depicted in Fig. 1a, primary EnSC were decidualized with a progestin (medroxyprogesterone acetate, MPA) and a cyclic adenosine monophosphate analogue (8-bromo-cAMP, cAMP) for 8 days. The differentiation signal was then withdrawn for 2 days to assess progesterone/cAMP-dependency of decidual subsets. Cells were recovered every 48 h and subjected to single-cell analysis using nanoliter droplet barcoding and high-throughput RNA-sequencing[34]. Approximately 800 cells were sequenced per timepoint, yielding on average 1282 genes per cell. After computational quality control (Supplementary Fig. 1), 4580 cells were assigned to 7 transcriptional cell states using Shared Nearest Neighbour (SNN) and t-Distributed Stochastic Neighbour Embedding (t-SNE) methods. Figure 1b shows cells colour-coded by day of treatment and Fig. 1c by transcriptional state (S1-7). The top 10 differentially expressed genes (DEG) between the 7 cell states are presented as a heatmap (Supplementary Fig. 2 & Supplementary Data 1). Apart from a discrete population of proliferative cells (S1), the bulk of undifferentiated EnSC were in S2. By day 2 of decidualization, most cells had transitioned to S3, which differed from S2 by 898 DEG (Supplementary Data 2). This precipitous transcriptomic response is in keeping with an acute cellular stress response and coincided with a transient rise in IL-6 and IL-8 secretion independent of transcription (Supplementary Fig. 3). Decidualizing cells then progressed in

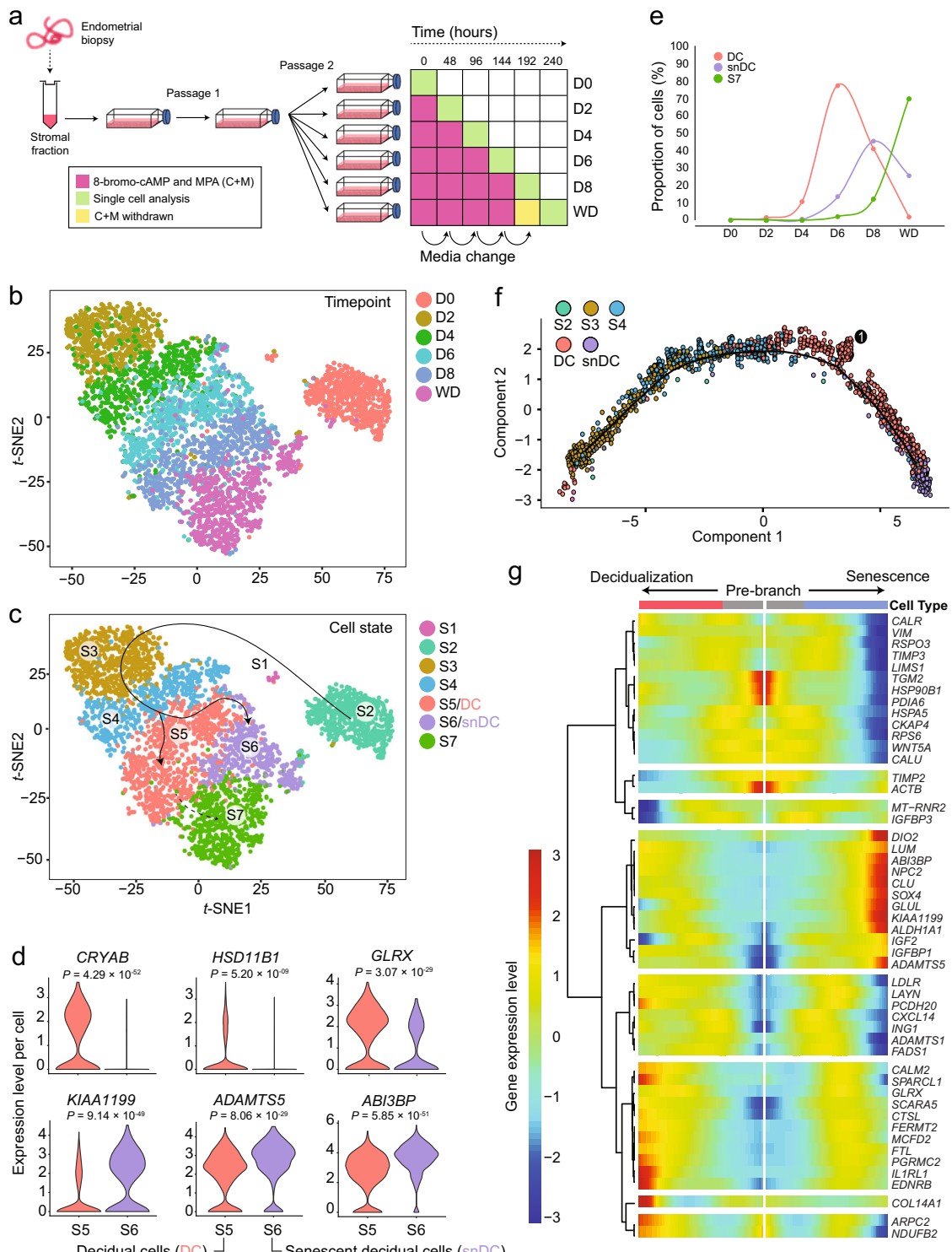

**Fig. 1 Single-cell reconstruction of the decidual pathway in culture. a** Schematic representation of in vitro time-course experiment. **b** *t*-SNE projection of 4580 EnSC, colour-coded according to days of decidualization (D0-8) and upon withdrawal (WD) of the differentiation signal for 48 h. D0 represents undifferentiated EnSC. **c** The same *t*-SNE plot now colour-coded according to transcriptional state (S1–7). **d** Violin plots showing log-transformed, normalized expression levels for indicated genes in decidual cells (DC, state S5) and senescent decidual cells (snDC, state S6). **e** Relative proportion of total cells assigned to states S5 (DC), S6 (snDC) and S7 at each experimental timepoint. **f** Decidualizing EnSC were placed in pseudotime to reconstruct the trajectory of differentiation, revealing a continuous trajectory towards senescence with a single branch point marking the divergence of DC. **g** Heatmap showing gene dynamics during cell state transition at the branch point shown in panel (**f**). Columns are points in pseudotime while rows represent the 50 most dynamic genes at the branch point. The beginning of pseudotime is in the middle of the heatmap and the trajectory towards DC and snDC are indicated by the arrows. Hierarchical clustering visualizes modules of genes with similar lineage-dependent expression patterns.

synchrony to S4 by day 4 after which they segregated into two transcriptionally distinct subsets, S5 and S6 (Fig. 1c). Analysis of independent EnSC cultures confirmed that decidualization polarizes cells into two subsets (Supplementary Fig. 4). Known decidual stress defence genes (e.g. CRYAB, HSD11B1 and GLRX)[35–37] were enriched in S5 (designated DC) whereas genes involved in oxidative stress signalling and cellular senescence, including KIAA1199, CLU and ABI3BP[38–41], prevailed in S6 (designated snDC) (Fig. 1d and Supplementary Fig. 5). Notably, on day 6, the proportion of DC and snDC was 78% and 13%, respectively (Fig. 1e and Supplementary Table 1). By day 8, snDC were almost as abundant as DC (42% and 45%, respectively), suggesting rampant secondary senescence. Withdrawal of the decidualization signal on day 8 elicited a further transcriptional change (S7), which was accounted for by partial de-differentiation of >95% of DC (Supplementary Data 2 and Supplementary Table 1). By contrast, snDC were much more refractory to withdrawal, suggesting loss of progesterone/cAMP-dependency.

To explore the 'switching' of DC into snDC further, we ordered decidualizing cells (D2-8) in pseudotime and colour-coded them by transcriptional state. As shown in Fig. 1f, this analysis revealed that decidualizing EnSC progress along a continuous trajectory towards senescence. This trajectory was interrupted by a single branch point, marking the divergence of DC and a subset of DC already transitioning towards a senescent phenotype. The top 50 genes underpinning this branch point are depicted in a modified heatmap and clustered hierarchically to visualize modules of genes with similar expression patterns (Fig. 1g). DIO2, coding iodothyronine deiodinase 2, was identified as a major branch gene in the senescent pathway. This enzyme catalyses the conversion of prohormone thyroxine (T4) into bioactive triiodothyronine (T3)[42], suggesting increased energy metabolism in snDC. Multiple genes coding for senescence-associated ECM remodelling factors were co-regulated with DIO2, including LUM (lumican)[43], CLU (clusterin)[41], ADAMTS5 (ADAM metallopeptidase with thrombospondin type 1 motif 5)[44], KIAA1199 (also known as cell migration inducing hyaluronidase 1, CEMIP)[39] and ABI3BP (ABI family member 3 binding protein)[38]. Notably, IGFBP1, a widely used decidual marker gene[8], was also part of this module. FTL and SCARA5, along with known decidual genes such as GLRX and IL1RL1, were part of a prominent branching module in the non-senescent decidual pathway (Fig. 1g). FTL encodes ferritin light chain (L-ferritin) and SCARA5 (scavenger receptor class A member 5) the L-ferritin receptor, suggesting increased iron storage and detoxification capacity in DC.

Taken together, the single-cell analysis confirmed that decidualization is a multistep process that starts with an acute stress response, which in turn synchronizes transition of cells through intermediate transcriptional states before emerging mainly as DC and some snDC. We also demonstrated that snDC rapidly perpetuate the senescent phenotype across the culture, resulting in chronic senescence and increased expression of ECM constituents and proteases and other SASP components.

**Co-regulated decidual gene networks.** We used k-means cluster analysis to identify networks of co-regulated genes across the decidual pathway (Supplementary Data 3). Analysis of 1748 DEG yielded 7 networks of uniquely co-regulated genes. Figure 2 depicts individual networks annotated for selected transcription factor (TF) genes with core roles in decidualization. Network A1 genes are rapidly downregulated within the first 48 h of the decidual process after which expression remains largely stable. The most notable TF to be 'reset' in this manner upon decidualization is the progesterone receptor (PGR). This observation is in keeping with a previous study purporting that overexpressed

PGR blocks the formation of multimeric transcriptional complexes upon decidualization by squelching key co-regulators[45]. Network A2 genes, including HOXA10, are progressively downregulated upon decidualization. Networks B1 and B2 comprise of biphasic genes that peak prior to the emergence of DC and snDC. They include several genes coding for pivotal decidual TFs, such as STAT3, MEIS1, KLF9, WT1 and FOXO1[8,19]. Genes in network C1 are gradually induced upon decidualization and then peak in DC. Prominent TFs in this network are heart and neural crest derivatives expressed 2 (HAND2) and FOS like 2 (FOSL2), a potent PGR co-regulator[46]. By contrast, two distinct networks underpinned the emergence of snDC. Network C2 genes rise gradually during the initial decidual phase but expression is then rapidly accelerated, especially in snDC. Interestingly, this network is enriched in gene ontology terms 'secreted' (Benjamini adjusted $P = 3.7 \times 10^{-5}$, modified Fisher Exact test) and 'type I interferon signalling pathway' (Benjamini adjusted $P = 1.4 \times 10^{-6}$, modified Fisher Exact test) (Supplementary Data 4), both canonical features of cellular senescence[24]. A prominent TF in this senescence-associated network is signal transducer and activator of transcription 1 (STAT1), which is not only activated by interferon signalling but is also a potent inhibitor of PGR signalling in endometrial cells[47]. A second biphasic network, designated network D, is also associated with snDC. Genes in this network are initially repressed upon decidualization and then re-expressed predominantly in snDC. Intriguingly, this network is enriched in genes known to be repressed by PGR, including the TF genes SOX4 and FOXP1[14]. Taken together, the data show that complex networks of co-regulated genes underpin progression of cells along the decidual pathway and suggest that altered expression levels of PGR co-regulators, such as FOSL2 and STAT1, may account for progesterone-dependency of DC and progesterone-resistance of snDC.

**Coordinated activation of immune surveillance genes.** We mined the networks further for genes encoding factors implicated in immune surveillance of senescent cells. CXCL14 and IL-15 drive uNK-cell chemotaxis and proliferation and activation, respectively[9,48]. TIMP-3 (TIMP metallopeptidase inhibitor 3) plays a critical role in immune recognition of stressed or senescent cells by inhibiting proteolytic cleavage of surface stress ligands needed for NK cell recognition[49]. Intriguingly, CXCL14, IL-15 and TIMP-3 belong to the same biphasic gene network (B2), characterized by peak expression immediately prior to the emergence of DC and snDC (Fig. 3a). Subsequently, expression of these genes drops markedly in snDC but much less so in DC. To explore this concept of 'programmed' immune surveillance further, we monitored the secreted levels of CXCL14, IL-15 and TIMP-3 every 48 h over an 8-day time-course in four independent decidualizing cultures. As shown in Fig. 3b, secreted levels of all 3 factors rise quickly during the initial decidual phase. While the levels of CXCL14 and TIMP-3 then appear to plateau, IL-15 continues to accumulate in the supernatant.

Clusterin (CLU) and the soluble IL-33 decoy receptor sST2 (encoded by IL1RL1) are putative secreted markers of snDC and DC, respectively (Fig. 1g). As shown in Fig. 3c, secreted levels of both markers increase markedly upon decidualization, although the rise in sST2 levels generally preceded CLU secretion. We speculated that uNK cells would selectively attenuate CLU secretion by targeting snDC. To test this hypothesis, uNK cells were isolated from the supernatants of freshly cultured EnSC using magnetic-activated cell sorting (MACS) and the purity and viability of cells confirmed by flow cytometry (Supplementary Fig. 6). The experimental design is depicted in Fig. 3d. To monitor uNK-cell killing of senescent cells, we measured

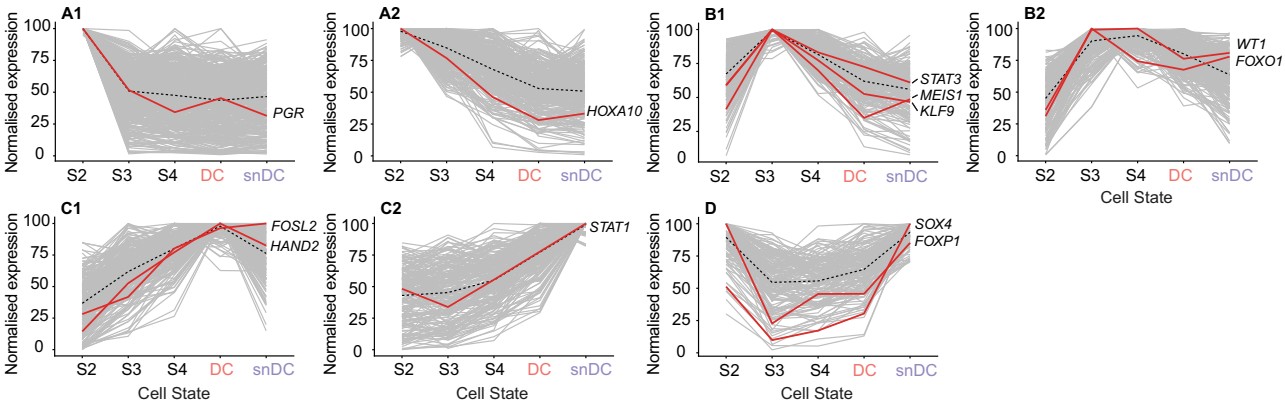

**Fig. 2 K-means cluster analysis identifies co-regulated decidual gene networks.** Analysis of 1748 DEG across the decidual pathway (D0–D8) yielded seven networks of uniquely co-regulated genes. Networks are annotated for selected transcription factor genes with core roles in decidualization.

senescence-associated β-galactosidase (SAβG) activity, a widely used senescence marker[26], in undifferentiated and decidualizing cultures with or without uNK cells. As reported previously[9], the increase in SAβG activity upon decidualization was entirely abrogated by co-cultured uNK cells (Fig. 3e). Further, uNK cells reduced the secreted levels of CLU (Fig. 3f), whereas the impact on sST2 levels was minimal (Fig. 3g). To confirm that uNK cells target snDC, six independent primary EnSC cultures were decidualized for 8 days in the presence or absence of dasatinib, a potent senolytic drug[9]. As shown in Supplementary Fig. 7, dasatinib also markedly reduced SAβG activity and selectively inhibited CLU whereas the effect on sST2 secretion was minimal, thus recapitulating the actions of uNK cells. Taken together, the data show that the expression and secretion of decidual factors involved in uNK-cell recruitment and activation is hardwired to coincide with the emergence of snDC. If recapitulated in vivo, these observations further suggest an important role for uNK cells in limiting the detrimental impact of snDC in early pregnancy.

**Single-cell analysis of peri-implantation endometrium.** Decidualization involves activation of numerous genes that are either constitutively expressed or induced in glandular epithelial cells following ovulation[8]. To identify stroma-specific marker genes of a divergent peri-implantation decidual response, we subjected freshly isolated endometrial cells to scRNA-seq. Biopsies were timed relative to the pre-ovulatory luteinizing hormone (LH) surge to coincide with the midluteal implantation window (LH + 8; $n = 3$) or the start of the late-luteal phase (LH + 10; $n = 3$). Following quality control (Supplementary Fig. 8), $t$-SNE analysis assigned 2847 cells to 5 clusters colour-coded to indicate the day of biopsy (Fig. 4a). Additional dimensionality reduction analysis was performed on immune cells (Fig. 4b). Clusters were designated based on canonical marker genes as endothelial cells (EC; $n = 141$), epithelial cells (EpC; $n = 395$), immune cells (IC; $n = 352$) and EnSC ($n = 1943$) (Fig. 4c and Supplementary Data 5). In addition, a discrete but as yet uncharacterized cluster of proliferating cells (PC; $n = 16$) was identified (Fig. 4a). EpC segregated in 4 clusters with the most abundant population (EpC1) expressing prototypic receptivity genes (e.g. *GPX3*, *PAEP* and *DPP4*) (Fig. 4c)[50]. EpC2 are ciliated epithelial cells found interspersed throughout endometrial glands (Supplementary Fig. 9). EpC3 were derived predominantly, but not exclusively, from a single biopsy whereas EpC4 represented an ambiguous population expressing both epithelial and stromal marker genes (Fig. 4c). uNK cells, representing 89% of all IC, clustered into three subpopulations (NK1-3; Fig. 4b). Notably, different NK populations have also recently been described in pregnant

decidua[51]. Based on CIBERSORT analysis, the remaining immune populations were identified as naive B-cells (IC1), monocytes (IC2), and macrophage/dendritic cells (IC3) (Fig. 4d and Supplementary Data 6). The abundance of various cell types and subsets in each biopsy is tabulated in Supplementary Table 1.

**Marker genes of diverging decidual states.** Next, we focused on the EnSC, which clustered prominently by day of biopsy in the $t$-SNE plot (Fig. 4a). Progression from LH + 8 to LH + 10 was associated with altered expression of 518 genes in EnSC (Supplementary Data 7), 49% of which are also part of the 7 co-regulated gene networks in vitro ($P < 10^{-143}$, hypergeometric test). Although our single-cell analysis was limited to six biopsies obtained on only two time-points, we surmised that the relationship between genes that drive the divergence of DC and snDC in vitro would, at least partly, be conserved in stromal cells in vivo. To test this hypothesis, we selected the top 50 genes of the branch point in the decidual time-course and determined the Pearson correlations for each gene pair in EnSC in vitro and in vivo on LH + 8 and LH + 10, respectively. We then calculated the sum of coefficients, ranging from +2 to −2. Congruency was defined as the sum of correlation coefficients of >1 or <−1 for positively and negatively co-regulated genes, respectively (Supplementary Fig. 10). Using this criterion, 27% of gene pairs were congruent on both LH + 8 and LH + 10, which is significantly more than expected by chance alone ($P < 10^{-30}$ and $P < 10^{-58}$, respectively; hypergeometric test).

We reasoned that informative marker genes of a divergent decidual response in vivo should be highly enriched in stromal cells, not regulated in glandular epithelium, and have a temporal profile across the luteal phase commensurate with the expected switch of DC to snDC prior to menstruation. Out of 50 branch genes, 5 genes (*TIMP-3*, *IGF2*, *DIO2*, *SCARA5* and *ABI3BP*) were highly enriched in EnSC compared with EpC, EC or IC. When cross-referenced against two publicly available datasets [Gene Expression Omnibus (GEO) ID: GSE84169 and GEO ID: GDS2052], only *SCARA5* and *DIO2* met all criteria. Notably, *SCARA5* belongs to the C1 network of genes whose expression peaks in DC whereas *DIO2*, a gene repressed by the activated PGR in decidualizing EnSC[14], belongs to the D network of senescence-associated genes (Fig. 5a). We examined the relative expression of both genes in different endometrial cell types (Fig. 5b) as well as their temporal expression across the cycle (Fig. 5c). Next, we measured *SCARA5* and *DIO2* transcript levels by RT-qPCR in 250 samples obtained across the implantation window (LH + 6–11) to generate percentile graphs based on the statistical distribution in gene expression for each day (Fig. 5d). To

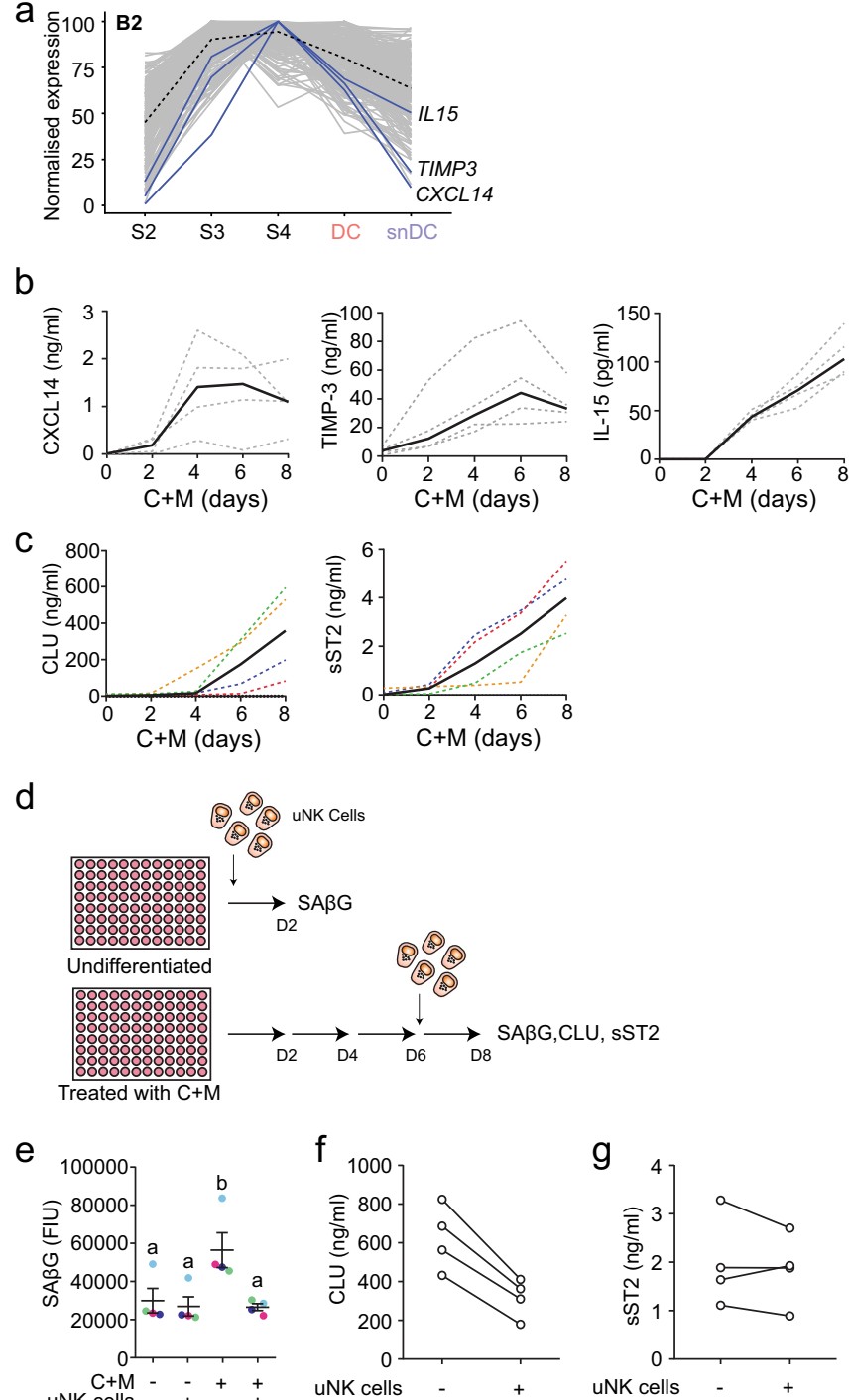

**Fig. 3 Coordinated expression of decidual immune surveillance genes. a** Decidual gene network B2 annotated to highlight genes implicated in uNK-cell activation and immune surveillance of senescence cells. **b** Primary EnSC cultures were decidualized with 8-bromo-cAMP and MPA (C+M) for the indicated days. ELISAs were performed on spent medium collected at 48 h intervals to examine secreted levels of CXCL14, IL-15 and TIMP-3. Grey dotted lines indicate secreted levels in individual cultures ($n = 4$). Black solid lines indicate the median level of secretion. **c** Four independent primary EnSC cultures were decidualized with 8-bromo-cAMP and MPA (C+M) for the indicated days. ELISAs were performed on spent medium collected at 48 h intervals to examine secreted levels of clusterin (CLU) and sST2. Cultures established from the same biopsy are colour matched between plots (dotted lines). Black solid lines indicate the median level of secretion. **d** Schematic representation of uNK-cell co-culture experiments. A total of 5000 primary uNK cells were co-cultured for 48 h from the indicated timepoint with 50,000 decidualized cells seeded in 96-well plates. Four independent EnSC cultures were used. **e** SAβG activity in undifferentiated and decidualized (C+M) cells co-cultured with or without uNK cells. Four independent EnSC cultures were used. **f** Secretion of CLU in decidualized (C+M) cells co-cultured with uNK cells relative to decidualized cells cultured without uNK cells. **g** Secreted sST2 levels in the same culture. Four independent EnSC cultures were used; lines connect paired samples.

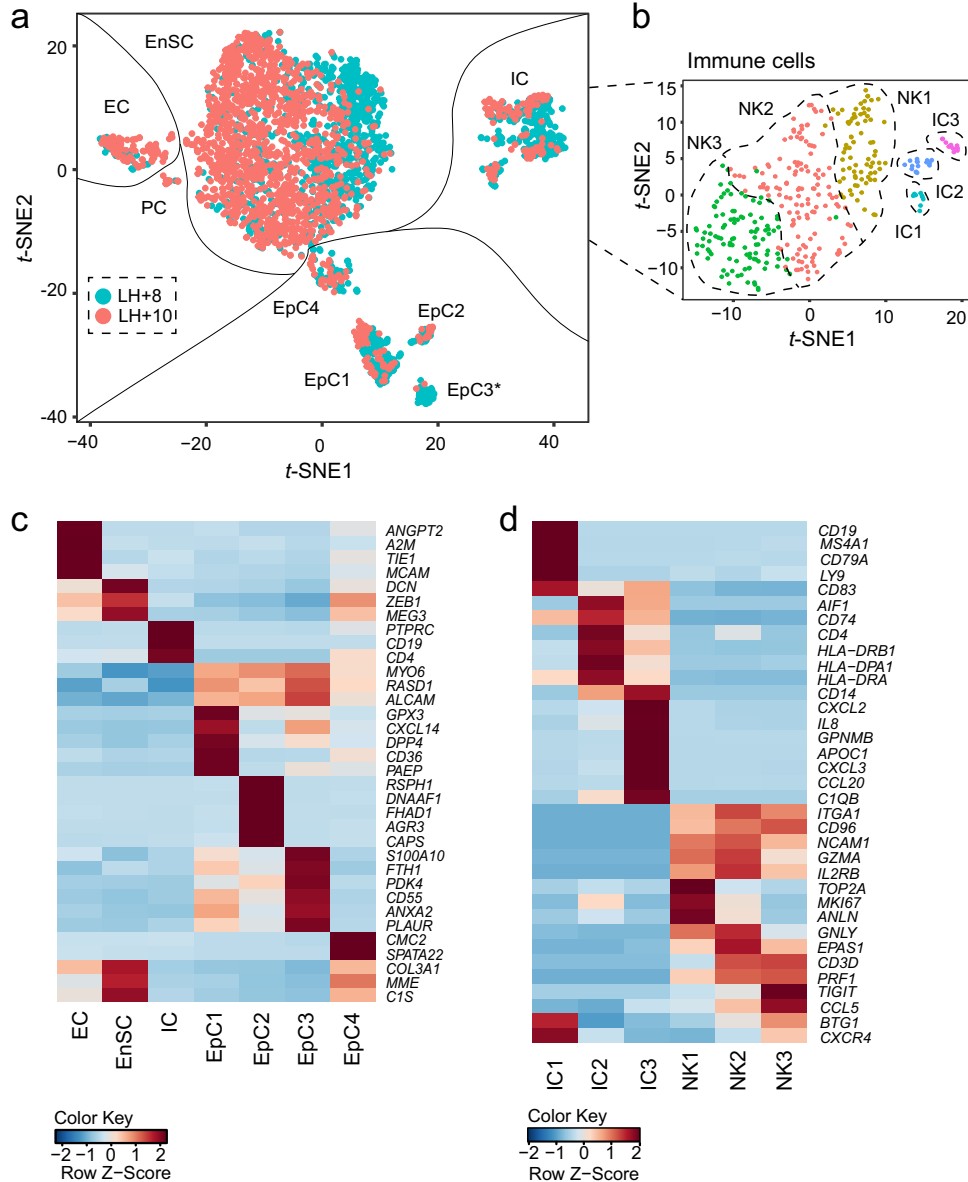

**Fig. 4 Identification of endometrial cell types and subsets during the implantation window. a** *t*-SNE plot of 2847 cells isolated from six LH-timed biopsies, coloured to indicate reported cycle day (LH + 8 or LH + 10), captures all major endometrial cell types, including epithelial cells (EpC), immune cells (IC), endothelial cells (EC), stromal cells (EnSC) and a discrete but transcriptionally distinct proliferative (PC) stromal subpopulation. EpC segregated in four subpopulations (EpC1-4). * indicates EpC3 contributed predominantly by a single sample. **b** additional dimensionality reduction separated immune cells into three uNK-cell subsets (NK1-3), naive B-cells (IC1), monocytes (IC2) and macrophage/dendritic cells (IC3). **c** Heatmap showing relative expression (z-score) of markers defining cell types and EpC subpopulations. *MYO6*, *RASD1* and *ALCAM* are included as pan-epithelial genes. **d** Heatmap showing relative expression of markers defining endometrial IC populations during the implantation window, including three uNK-cell subsets.

determine if *SCARA5* and *DIO2* transcripts are co-expressed or mark different decidual cells, we performed multiplexed single-molecule in situ hybridization (smISH) on endometrial biopsies obtained on the same cycle day but deemed *SCARA5*$^{HIGH}$/ *DIO2*$^{LOW}$, *SCARA5*$^{LOW}$/*DIO2*$^{HIGH}$ or *SCARA5*$^{AVERAGE}$/ *DIO2*$^{AVERAGE}$ based on the corresponding percentile graphs (Fig. 5e). Most EnSC were *SCARA5*$^+$ but *DIO2*$^−$ in *SCARA5*$^{HIGH}$/ *DIO2*$^{LOW}$ biopsies whereas the opposite pattern was observed in *SCARA5*$^{LOW}$/*DIO2*$^{HIGH}$ samples. However, *SCARA5*$^{AVERAGE}$/ *DIO2*$^{AVERAGE}$ samples consisted of mixture of *SCARA5*$^+$ or *DIO2*$^+$ cells as well as intermediate cells expressing both transcripts. Finally, we used a computational approach to isolate EnSC relatively enriched in *SCARA5* transcripts but reduced *DIO2* expression and vice versa. As shown in Supplementary

Fig. 11, *SCARA5*$^{enriched}$/*DIO2*$^{reduced}$ cells are characterized by the expression of multiple decidual TF and stress-resistance genes whereas a stress gene signature was prominent in *SCARA5*$^{reduced}$/ *DIO2*$^{enriched}$ cells (Supplementary Data 8).

**RPL is associated with an aberrant decidual response**. To examine if an aberrant decidual response in cycling endometrium is linked to increased risk of pregnancy loss, we analysed LH-timed endometrial biopsies from 90 control subjects and 89 RPL patients. Demographic and clinical characteristics are presented in Table 1. Each biopsy was divided and processed for RT-qPCR analysis and immunohistochemistry. *SCARA5* and *DIO2* transcript levels were measured and normalized for the day of the

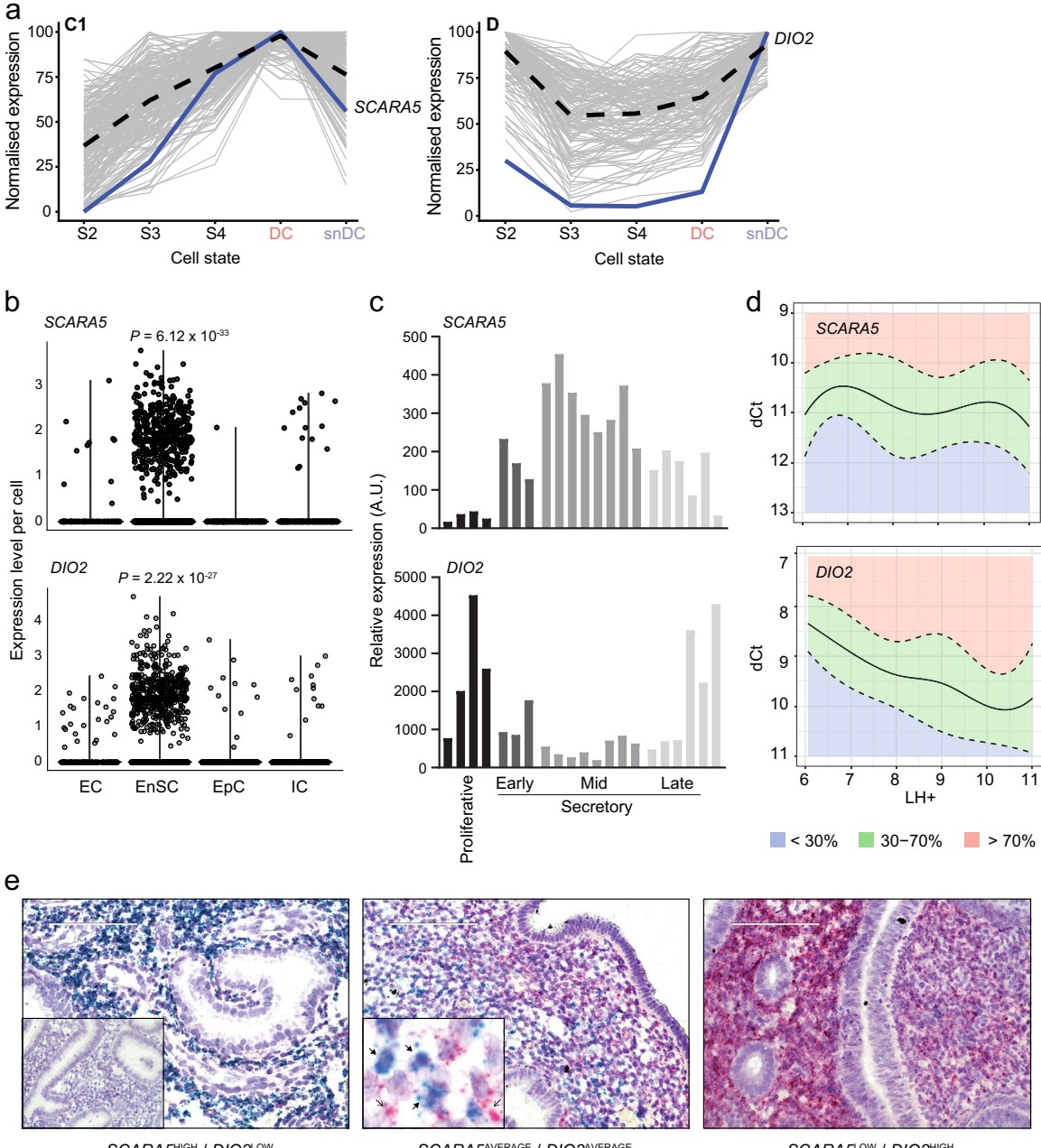

**Fig. 5 SCARA5 and DIO2 as marker genes for divergent decidual states. a** *SCARA5* and *DIO2* belong to two distinct decidual gene networks with peak expression in DC and snDC, respectively. **b, c** Spatial and temporal expression of *SCARA5* and *DIO2* in cycling human endometrium: **b** violin plots showing expression of *SCARA5* and *DIO2* in vivo in EnSC, EC (endothelial cells), EpC (epithelial cells) and IC (immune cells) (Wilcoxon rank sum test with Bonferroni correction); **c** expression of *SCARA5* and *DIO2* in proliferative and early-, mid-, late-luteal phase endometrium. Each bar represents an individual biopsy. The data were retrieved from microarray data deposited in the Gene Expression Omnibus (GEO Profiles, GDS2052). **d** *SCARA5* and *DIO2* transcript levels quantified by RT-qPCR analysis in 250 endometrial biopsies obtained between LH + 6 and LH + 11. Centile calculations were performed on dCt values using R software and graphs were generated based on the distribution of expression on each day. The median number of samples for each day was 43 (range: 30–46). **e** Multiplexed single-molecule in situ hybridization (smISH) on three endometrial biopsies (LH + 10) deemed *SCARA5*HIGH/*DIO2*LOW (95th and 20th percentile, respectively), *SCARA5*AVERAGE/*DIO2*AVERAGE (60th and 52nd percentile, respectively), and *SCARA5*LOW/*DIO2*HIGH (4th and 91st percentile, respectively). Insert in the left panel shows hybridization with a negative control probe. Insert in middle panel shows *SCARA5*+ cells (blue, closed arrows) and *DIO2*+ cells (pink, open arrows) in close proximity. Scale bar: 100 µM. Original magnification: ×40.

cycle based on their respective percentile graphs (Fig. 5d). Immunohistochemistry was used to quantify uNK cells using a standardized and validated approach that involves measuring the number of CD56+ uNK cells per 100 cells in proximity of the luminal epithelium[52]. The uNK-cell data were then normalized for the day of the biopsy in the cycle as reported previously[9]. As

shown in Fig. 6a, lower *SCARA5* but higher *DIO2* percentiles in RPL samples indicated a diverging pro-senescent decidual response. Further, uNK cells were significantly less abundant in RPL compared with control subjects (*P* < 0.0001, Welch two-sided *t*-test). We reasoned that *SCARA5*+ DC and uNK cells drive successful transformation of the stroma into the decidua of

**Table 1 Subject demographics.**

| | Control[a] ($n = 90$) | RPL[b] ($n = 89$) | P-value[c] |
|---|---|---|---|
| Age (years) (median ± IQR) | 36 (33–37) | 36 (33–38) | 0.253 |
| BMI (median ± IQR) | 22 (21–25) | 26 (22–30) | <0.0001 |
| LH + day (median ± IQR) | 9 (7–10) | 9 (7–10) | 0.968 |
| First trimester loss [median (range)] | 0 (0–2) | 5 (3–18) | <0.0001 |
| Live births [mean (range)] | 0 (0–2) | 0.35 (0–2) | <0.0001 |

[a]Control subjects were women awaiting IVF treatment for a variety of reasons, including male-factor, unexplained, and tubo-ovarian infertility. All subjects had regular cycles and considered to have good prognosis; recurrent IVF failure (RIF: >3 consecutive IVF failures with good quality embryos) patients were excluded
[b]All Recurrent pregnancy loss (RPL) patients in this cohort had three or more consecutive miscarriages
[c]Data were tested for normality using Shapiro–Wilk test. P-value was calculated by two-tailed unpaired student's t-test for normally distributed data (age) or two-tailed Mann Whitney test for non-normally distributed data (BMI, LH+, first trimester loss and live births)

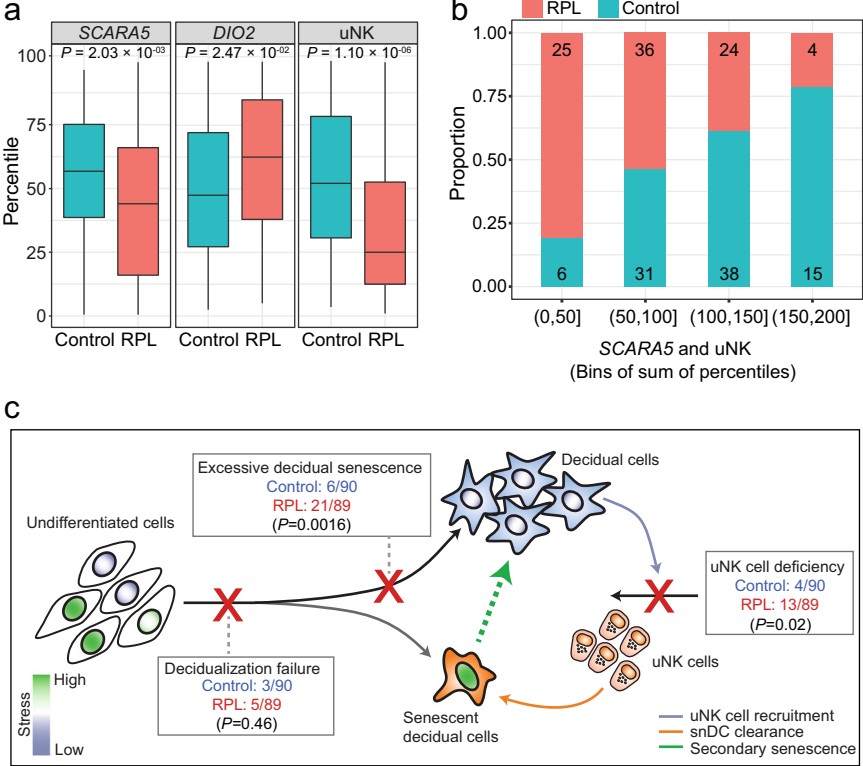

**Fig. 6 Impaired fate divergence of decidual cells in RPL. a** Distribution of uNK cells and *SCARA5* and *DIO2* percentiles in timed endometrial biopsies of control subjects ($n = 90$) and RPL patients ($n = 89$) (Welch two-sided t-test). **b** Number of RPL and control subjects across four bins defined by the sum of *SCARA5* and uNK-cell percentiles. **c** Diagram illustrating the decidual pathway. Fate divergence of EnSC upon decidualization relates to the level of replicative stress (indicating by nuclear shading) incurred by individual cells during the proliferative phase. Stress-resistant decidual cells recruit and activate uNK cells to eliminate stressed/senescent decidual cells through granule exocytosis. Different defects along the decidual pathway can be identified, including 'decidualization failure', 'excessive decidual senescence' and 'uNK cell deficiency'. The frequency of each defect in RPL and control subjects is shown ($\chi^2$ test).

pregnancy. Hence, we created four bins based on the sum of *SCARA5* and uNK percentiles and determined the number of RPL and control subjects in each bin. As shown in Fig. 6b, 81% (25/31) of subjects assigned to the lowest bin were RPL patients whereas 79% (15/19) in the highest bin were control subjects ($P < 0.0001$; Fisher's exact test). Our analysis also enabled a definition of putative defects across the pathway, including 'decidualization failure' (*SCARA5*: ≤30th percentile, *DIO2*: ≤30th percentile), 'excessive decidual senescence' (*SCARA5*: ≤30th percentile, *DIO2*: ≥70th percentile), and 'uNK cell deficiency' (uNK cells: ≤30th percentile, *SCARA5* and *DIO2*: >30th–<70th percentile) (Fig. 6c). Excessive decidual senescence and uNK-cell deficiency were significantly more common in RPL patients compared to control subjects ($P = 0.0016$ and $P = 0.02$, respectively; $\chi^2$ test). Taken together, the incidence of an aberrant decidual response (i.e. encompassing all defects) was approximately three times higher in RPL compared to control subjects (44% versus 14%, respectively, $P = 0.00013$; Fisher's exact test).

## Discussion

In menstruating species, including humans, decidualization is not under the control of an implanting embryo but initiated during the midluteal phase of each cycle[53]. Once triggered, the decidualizing endometrium becomes inextricably dependent on sustained progesterone signalling. In the absence of implantation, falling progesterone levels elicit an inflammatory decidual response which, upon recruitment and activation of leukocytes,

leads to tissue breakdown and menstrual shedding[54,55]. Thus, a critical challenge at implantation is to simultaneously avoid imminent endometrial breakdown while transforming the cycling endometrium into a semi-permanent tissue, the decidua, maintained throughout pregnancy. We recently argued that successful transformation of the endometrium in early pregnancy requires both successful differentiation of EnSC into DC and prompt elimination of snDC by activated uNK cells; and conversely, that defects in this process predisposes for early pregnancy loss[9]. To test our hypothesis, we set out to identify putative marker genes of an aberrant decidual response using single-cell RNA-seq.

We first generated a detailed transcriptional map of primary EnSC decidualized in culture over 8 days followed by withdrawal of the differentiation signal for 2 days. This analysis revealed a multistep process that starts with a precipitous transcriptional response in differentiating EnSC, which is followed by synchronous transition of cells through intermediate states and then the emergence of DC and some snDC. A recent study asserted that DC emerged in evolution from rewiring of an ancestral cellular stress response[19]. This conclusion was based on the observation that endometrial fibroblasts isolated from marsupials, which diverted from eutherians 60 to 80 million years ago[56], activate similar core regulatory genes as human EnSC in response to a decidualizing stimulus and then mount an acute stress response akin to the initial decidual phase[19]. The key difference is that most EnSC emerge from this process as DC expressing multiple anti-inflammatory and stress-resistance genes. However, some EnSC emerge as snDC, thus recapitulating the ancestral response. We also demonstrated that snDC rapidly convert DC into secondary senescent cells, a process abrogated by co-cultured uNK cells. Importantly, secondary senescence had a dramatic impact on decidual gene expression, characterized by marked upregulation of genes encoding for ECM proteins and proteases and other SASP components. The mechanisms driving secondary senescence of DC warrant further exploration, including the possibility that EnSC become sensitized to juxtacrine senescence upon differentiation into DC. A hallmark of snDC is progesterone-resistance, exemplified by expression of PGR-repressed genes, such as SOX4 and FOXP1,[14] and lack of responsiveness to progesterone and cAMP withdrawal. Analysis of co-regulated gene networks indicated that a switch in PGR co-regulators from FOSL2 to STAT1 may account for progesterone-resistance in snDC, although this conjecture will need to be tested experimentally. Apart from generating a temporal map of TF and effector genes underpinning progression of cells along the decidual pathway, the network analysis also revealed that multiple genes involved in immune surveillance of senescent cells, including CXCL14, IL-15 and TIMP-3, are hardwired to be activated prior to the emergence of snDC. Interestingly, human chorionic gonadotrophin has been shown to stimulate uNK-cell proliferation directly[57], which suggests potential cooperation between the implanting embryo and DC in eliminating snDC.

Our in vitro analysis suggested that the default trajectory of decidualizing EnSC is towards cellular senescence, which can only be avoided by timely clearance of snDC by uNK cells. Failure to do so leads to chronic senescence, at least in culture. There are multiple mechanisms by which chronic senescent cells promote tissue dysfunction, including perturbation of stem cells, disruption of ECM, induction of tissue inflammation, and propagation of senescence in neighbouring cells[25,27,32]. When extrapolated to the in vivo situation, these observations imply that a pro-senescent decidual response during the peri-implantation window would lead to the formation of an intrinsically unstable decidual–placental interface in pregnancy. To test this hypothesis, we set out to identify stroma-specific marker genes to discriminate between a physiological and pathological decidual

response. Single-cell transcriptomic analysis of midluteal endometrium yielded a number of notable results, including the discovery of a discrete but as yet uncharacterized population of proliferative mesenchymal cells, the characterization of different epithelial cell subsets, and the identification of three uNK-cell states defined in part by the abundance of cell-cycle genes. For example, the NK1 population, representing the most proliferating uNK cells, abundantly express genes involved in granule exocytosis (e.g. PRF1, GNLY, GZMA and GZMB). By contrast, NK3 cells express low levels of cell-cycle genes but are defined by CCL5 and CXCR4 expression. Notably, CXCR4$^+$ uNK cells have previously been implicated in vascular remodelling in pregnancy[58].

As the in vivo single-cell analysis was restricted to few samples across two time-points, we restricted our search for putative marker genes to the 50 top in vitro branching genes that marked divergence of DC and DC already transitioning towards senescence. Out of 50 genes, SCARA5 and DIO2 were deemed informative marker genes of DC and snDC, respectively, based on their temporal regulation across the cycle in whole endometrial biopsies, lack of regulation in glandular epithelium, and level of enrichment in stromal cells. Further, multiplexed smISH showed that SCARA5$^+$ cells are distinct from DIO2$^+$ cells, which was further corroborated by the different transcriptomic profiles of SCARA5$^{enriched}$/DIO2$^{reduced}$ and SCARA5$^{reduced}$/DIO2$^{enriched}$ EnSC in the in vivo single-cell data set. To test our supposition that an aberrant decidual trajectory predisposes to pregnancy loss, we first generated percentile graphs for SCARA5 and DIO2 expression using 250 midluteal biopsies (LH + 6–11). This approach enables comparison of the relative level of gene expression in biopsies obtained at different days of the cycle. The relative abundance of uNK cells was also determined using a previously established percentile graph based on analysis of 1997 biopsies[9]. Next, we quantified the abundance of uNK cells and the expression of SCARA5 and DIO2 in timed endometrial biopsies from RPL and control subjects. Sample selection was based solely on reproductive history; all RPL patients had three or more consecutive miscarriages whereas control subjects were women with male-factor, tubal or unexplained infertility awaiting IVF treatment. Overall, our analysis indicated that pre-pregnancy endometrium in RPL is characterized by uNK-cell deficiency in parallel with a shift from a preponderance of DC to snDC. Next, we examined the frequency of different putative defects along the decidual pathway. Based on pre-specified but arbitrary criteria, we found that RPL is associated with 'excessive decidual senescence' and 'uNK cell deficiency' but not 'decidualization failure'. While promising, prospective studies are needed to define prognostic criteria that could be exploited for pre-pregnancy screening of women at risk of RPL.

Decidualization is an iterative process in cycling endometrium. At present, it is not clear if an aberrant decidual response persists from cycle-to-cycle or whether it represents an intermittent defect that affects some but not all cycles. The relative high cumulative live-birth rate in RPL favours the latter scenario[59]. Two distinct regulatory mechanisms control cellular homoeostasis in cycling endometrium. On the one hand, uNK cells may engender selective elimination of snDC during the midluteal phase, de facto rejuvenating the endometrium at the time of embryo implantation[9]. On the other, the cycling endometrium actively recruits bone marrow-derived cells (BDMC) capable of differentiating into stromal, epithelial and endothelial cells[60]. Experimental studies in mice have shown that BDMC enhance the regenerative capacity of non-pregnant endometrium[60], and contribute to rapid decidual expansion in pregnancy[61]. Interestingly, we recently reported a double-blind, randomised, placebo-controlled pilot trial demonstrating that sitagliptin, an oral antidiabetic drug and DPP4 inhibitor, is effective in increasing engraftment of BMDC

in the endometrium and attenuating the pro-senescent decidual response in RPL patients[62].

In summary, based on single-cell transcriptomic analysis, we propose that decidualization is a multistep process that ultimately leads to chronic senescence, a cellular state incompatible with the formation of a functional decidual–placental interface. Our data suggest that DC must engage uNK cells to eliminate snDC to escape this default pathway, although additional in vivo studies are required to substantiate this conjecture. Nevertheless, we identified *SCARA5* and *DIO2* as selective marker genes for DC and progesterone-resistant snDC, respectively. When combined with uNK cells, these marker genes can be used to map putative defects along the decidual pathway in cycling endometrium. Our findings raise the possibility of a simple screening test to identify women at risk of miscarriage and to monitor the effectiveness of pre-pregnancy interventions.

## Methods

**Ethical approval and sample collection**. The study was approved by the NHS National Research Ethics-Hammersmith and Queen Charlotte's & Chelsea Research Ethics Committee (1997/5065). All samples were obtained with written informed consent and in accordance with The Declaration of Helsinki (2000) guidelines. Human endometrial biopsies were obtained from women attending the Implantation Clinic, a dedicated research clinic at University Hospitals Coventry and Warwickshire (UHCW) National Health Service Trust. Surplus tissue from endometrial biopsies obtained for diagnostic purposes at the Implantation Research Clinic was used for this study. Samples were obtained during the luteal phase of ovulatory, non-hormonally stimulated menstrual cycles, timed relative to the pre-ovulatory LH surge, using a Wallach Endocell™ endometrial sampler. Overt uterine pathology was excluded by transvaginal ultrasound scan prior to the biopsy. Control subjects were women awaiting IVF treatment for a variety of reasons, including male-factor, unexplained, and tubo-ovarian infertility. Control subjects were considered to have good prognosis; patients with recurrent implantation failure, defined as three or more consecutive IVF failures with good quality embryos, were excluded. Recurrent pregnancy loss (RPL) patients had three or more consecutive miscarriages. Demographic details are presented in Table 1.

**Primary endometrial stromal cell (EnSC) culture**. Endometrial biopsies were collected in DMEM-F12 media supplemented with 10% dextran-coated charcoal-stripped FBS (DCC) and processed for primary EnSC culture as described[9]. Briefly, tissue was minced for 5 min before digestion with Collagenase Type Ia (500 µg/ml) and DNase I (100 µg/ml) for 1 h at 37 °C. Cell suspension was then filtered through a 40 µM cell sieve to remove undigested glandular cell clumps. The flow-through was cultured in DMEM/F12 with phenol red containing 10% DCC, 1X Antibiotic-Antimycotic mix, L-glutamine, estradiol and insulin (growth medium). Cells were cultured on tissue culture-treated plastic with media changed every 48 h. Cells were passaged at sub-confluence by 5 min treatment with 0.25% Trypsin-EDTA. For decidualization studies, confluent monolayers of human endometrial stromal cells (EnSC) at passage 2 were incubated overnight at 37 °C with 5% $CO_2$ in phenol red-free DMEM/F-12 containing 2% DCC, antibiotic/antimycotic and L-glutamine (2% media). To induce differentiation, cells were treated with 0.5 mM 8-bromo-cAMP (Sigma-Aldrich, Poole, UK) and 1 µM medroxyprogesterone acetate (MPA; Sigma-Aldrich) for the indicated time-points. For inhibition of senescence, cells were treated with 250 nM dasatinib (CST, Leiden, The Netherlands) throughout the differentiation time-course.

**Droplet generation and single-cell sequencing (Drop-Seq)**. Single-cell transcriptomes were captured in aqueous droplets containing barcoded beads using a microfluidic system (scRNA-seq: Dolomite Bio, Royston, UK) according to the manufacturer's protocol and based on the Drop-Seq method described by Macosko and colleagues[34]. Briefly, cells in suspension were placed into the remote chamber of the scRNA-seq system. Barcoded beads (Barcoded Bead SeqB; Chemgenes Corp., USA) in lysis buffer at a concentration of 280 beads/µl were loaded into the sample loop. Cell and bead solutions were run at a flow rate of 30 µl/min into a fluorophilic glass microfluidic chip with 100 µm etch depth (Single Cell RNA-seq Chip, Dolomite Bio) with droplet generation oil (Bio-Rad Laboratories, UK) at a flow rate of 200 µl/min for 15–18 min. Droplets were collected into a 50 ml Falcon tube, quality checked using a C-Chip Fuchs-Rosenthal Haemocytometer (Labtech, Heathfield, UK), and bead doublets counted. Droplet breakage, bead isolation and reverse transcription were performed exactly as described by Macosko and Goldman (Drop-Seq Laboratory Protocol version 3.1, http://mccarrolllab.org/download/905/) on 8000 beads per reaction, with two reactions per sample, to give ~800 single-cell transcriptomes attached to microparticles (STAMPS) per timepoint. PCR conditions were as per Macosko and Goldman. Clean-up with Agencourt AMPure XP beads (Beckman Coulter, High Wycombe, UK) was performed according to standard Illumina RNA-seq protocols with a 0.6× beads to sample ratio. cDNA was

eluted in 12 µl nuclease-free water and quality and size assessed using an Agilent Bioanalyzer High Sensitivity DNA chip (Agilent, Stockport, UK). For tagmentation 600 pg cDNA, as determined by Qubit High Sensitivity DNA assay (ThermoFisher, Paisley, UK), was processed according to Macosko and Goldman using Illumina Nextera XT DNA Sample Kit and Indexing Kit (Illumina, Cambridge, UK). Tagmented libraries were cleaned up using AMPure XP beads as before, with a 0.6× beads ratio followed by a repeat clean-up using 1× beads. Eluted libraries were analysed using Agilent Bioanalyzer High Sensitivity DNA chip to assess quality and determine library size, and concentration was determined by Qubit High Sensitivity DNA assay. Library dilution and denaturation was performed as per standard Illumina protocols and sequenced using NextSeq High Output 75 cycle V2 kit.

**Drop-Seq data alignment and quantification**. Initial Drop-Seq data processing was performed using Drop-Seq_tools-1.0.1 following the protocol described by Nemesh (seqAlignmentCookbook_v1.2Jan2016.pdf, http://mccarrolllab.com/dropseq). Briefly, reads with low-quality bases in either cell or molecular barcode were filtered and trimmed for contaminating primer or poly-A sequence. Sequencing errors in barcodes were inferred and corrected, as implemented by Drop-Seq_tools-1.0.1. Reads were aligned to the hg19 (Human) reference genome concatenated with ERCC annotations using STAR-2.5.3a[63], with the Gencode21 (Human) as reference transcriptome. Uniquely mapped reads, with ≤1 insertion or deletion, were used in quantification. Finally, the DigitalExpression tool[34] was used to obtain the digital gene expression matrix for each sample. Cell numbers were selected computationally from the inflection point in a cumulative distribution of reads plotted against the cell barcodes ordered by descending number of reads[64]. Cell barcodes beyond the inflection point are believed to represent 'ambient RNA' (e.g. contaminating RNA from damaged cells), not cellular transcriptomes, and therefore excluded from further analysis. This resulted in ~800 cells per timepoint, matching the number anticipated from processed bead counts.

**Cell aggregation analysis**. Analysis of DGE data was performed with Seurat v2[65]. To select high-quality data for analysis, cells were included when at least 200 genes were detected, while genes were included if they were detected in at least three cells. Cells which had more than 4500 genes were excluded from the analysis as were cells with more than 5% mitochondrial gene transcripts to minimize doublets and low-quality (broken or damaged) cells, respectively. After scaling and normalization of the raw counts in the DGE matrix, cell-cycle regression was applied. For cell aggregation, a set of highly variable genes was first identified, with an average expression mean between 0.0125 and 3 and a Log Variant to Mean Ratio of at least 0.5, which were used to perform principal component (PC) analysis. Based on statistical significance and the robustness of the results, the first 10 PCs were subsequently used as inputs for clustering via shared nearest neighbour (SNN) and subsequent *t*-distributed stochastic neighbour embedding (*t*-SNE) representation. The Seurat function 'FindAllMarkers' employing the Wilcoxon test was used to identify marker genes for each cell state cluster in the *t*-SNE representation. To obtain independent estimation of the number of unique cell types we used SC3 v1.3.18 (5), applying a consensus strategy and Tracy-Widom theory on random matrices to estimate the optimal number of clusters (*k*).

**Co-regulated gene networks**. *K*-means cluster analysis was performed in MeV v4.8 (6) to group marker genes based on co-expression across cell state clusters. Figure-of-merit (FOM) was run first to determine the number of expression patterns (*k*). The predictive power of the *k*-means algorithm was estimated using an FOM values for *k* from 1 to 20. *K*-means clustering was run using Pearson correlation metric for a maximum of 50 iterations. Gene ontology analysis was performed on the clustered genes using DAVID[66].

**Trajectory analysis**. Trajectory analysis was performed with Monocle v2.6.1[67]. Branch-point genes were identified with Branched Expression Analysis Modeling (BEAM) function.

**Gene expression correlation**. To test gene expression correlation between pairs of genes, expression was imputed for every cell using Markov Affinity-based Graph Imputation of Cells (MAGIC)[68]. Pearson correlation coefficients were calculated for top 50 genes determining the different trajectories at informative branch-points. To assess congruency between the time-course and biopsy datasets, the correlation coefficients were added resulting in sum of coefficients between −2 and +2. To infer gene expression associated with a pro-senescent decidual response, *SCARA5*^enriched/*DIO2*^reduced and *SCARA5*^reduced/*DIO2*^enriched EnSCs were selected after MAGIC imputation using the midpoint of the expression levels as thresholds. Differential gene expression on the selected cells was determined using the Seurat function 'FindMarkers' employing the Wilcoxon test, with GO analysis performed using DAVID[66].

**Drop-Seq analysis of timed endometrial biopsies**. Six LH-timed endometrial biopsies were processed as described in detail above and elsewhere[9]. After tissue digestion, red blood cells were removed from the flow-through by Ficoll density gradient centrifugation (ref. [9]). Single-cell fractions were then subjected to

**Drop-Seq analysis**. The time from biopsy in the clinic to cDNA synthesis was <4 h for all samples. Anonymized endometrial biopsies were obtained from women aged between 31 and 42 years with regular cycles, body mass index between 23 and 32 kg/m$^2$, and absence of uterine pathology on transvaginal ultrasound examination.

**Reverse transcription quantitative PCR (RT-qPCR)**. RNA was extracted from endometrial biopsies which had been snap frozen in clinic (<1 min after collection), using STAT-60 (AMS Biotechnology, Oxford, UK) according to the manufacturer's instructions. Reverse transcription was performed from 1 μg RNA using the Quantitect Reverse Transcription Kit (QIAGEN, Manchester, UK) and cDNA was diluted to 10 ng/μl before use in qPCR. Amplification was performed on a 7500 Real-Time PCR system (Applied Biosystems, Paisley, UK) in 20 μl reactions using 2 × PrecisionPlus Mastermix with SYBR Green and low ROX (PrimerDesign, Southampton, UK), with 300 nM each of forward and reverse primers. *L19* was used as a reference control. Primer sequences were as follows: *SCARA5* forward: 5′-CATGCG TGG GTT CAA AGG TG-3′, *SCARA5* reverse: 5′-CCA TTC ACC AGG CGG ATC AT-3′; *DIO2* forward: 5′-ACT CGG TCA TTC TGC TCA A-3′, *DIO2* reverse: 5′-TTC CAG ACG CAG CGC AGT-3′, *L19* forward: 5′-GCG GAA GGG TAC AGC CAA T-3′, L19 reverse: 5′-GCA GCC GGC GCA AA-3′. Centile calculations were performed on dCt values using R v3.5 software.

**Multiplexed single-molecule in situ hybridization**. Formalin-fixed paraffin-embedded (FFPE) samples were cut to 5 μm sections. RNA in situ hybridization was carried with RNAscope® 2.5 HD Duplex Reagent Kit (ACD, California, USA) with probes for SCARA5 (574781-C1) and DIO2 (562211-C2) according to manufacturer's guidelines. Following hybridization and amplification, slides were counterstained with 50% haematoxylin. Images were obtained using an EVOS AUTO microscope (ThermoFisher Scientific) with a ×40 objective lens.

**Isolation and culture of uNK cells**. Primary uNK cells were isolated from luteal phase endometrial biopsies as described previously[9]. Briefly, supernatant from freshly digested EnSC cultures was collected while uNK cells were isolated by magnetic-activated cell separation (MACS; Miltenyi Biotec, Bergisch Gladbach, Germany) using phycoerythrin (PE)-conjugated anti-CD56 antibody (Bio-Legend, San Diego, CA, USA), as per manufacturer's instructions. The CD56$^+$ positive fraction was collected by centrifugation and cultured in suspension for up to 5 days in RPMI media (Sigma-Aldrich) supplemented with 10% DCC-FBS, 1× Antibiotic-Antimycotic, and 2 ng/ml IL-15 (Sigma-Aldrich) to aid uNK-cell maturation. To increase yield, uNK cells from 3 to 5 subjects were pooled. For co-culture experiments, uNK cells were pelleted and re-suspended in DMEM/F-12 containing 2% DCC without IL-15. A total of 5000 uNK cells were added to 50,000 EnSC, decidualized or not, per well of a 96-well plate. MACS isolated uNK cells were analysed by flow cytometry after labelling with BD Horizon™ Fixable Viability Stain (FVS) 660 (BD Biosciences) in 1 × PBS (1:1000) and PE-conjugated CD56 (BD Biosciences; clone: B159; catalogue no: 555516; 1:5) according to manufacturer's instructions. Red blood cells (RBC) were removed from the negative fraction using 1×RBC lysis solution (BD Biosciences, catalogue number: 349202) for 10 min at room temperature. Samples were analysed by BD FACSMelody™ Cell Sorter (BD Biosciences) and data analysis was performed using FlowJo v10.

**Quantitative analysis of uNK cells in endometrial biopsies**. FFPE tissue sections were stained for CD56 (a uNK-cell-specific cell surface antigen) using a 1:200 dilution of concentrated CD56 antibody (NCL-L-CD56-504, Novocastra, Leica BioSystems). Stained slides were de-hydrated, cleared and cover-slipped in a Tissue-Tek® Prisma® Automated Slide Stainer, model 6134 (Sakura Flinetek Inc., CA, USA) using DPX coverslip mount. Bright-field images were obtained on a Mirax Midi slide scanner using a ×20 objective lens and opened in Panoramic Viewer v1.15.4 (3DHISTECH Ltd, Budapest, Hungary) for analysis. To avoid inconsistencies that reflect reduced uNK-cell densities at greater endometrial depths, CD56$^+$ cells were quantified in compartments directly underlying the luminal epithelium. Here, three randomly selected areas of interest for each biopsy were identified and captured within Panoramic Viewer before analysis in ImageJ image analysis software. Both luminal and glandular epithelial cells were removed manually before colour deconvolution into constituent brown (CD56 + staining) and blue (hematoxylin staining – stromal cells). The area of positive staining above a manually determined background threshold was used to quantify staining intensity. The uNK-cell percentage was calculated as the number of CD56$^+$ cells per 100 stromal cells and averaged from three images[9].

**Enzyme-linked immunosorbent assay (ELISA)**. Culture supernatants were collected every 2 days and centrifuged to clear cell debris prior to storage at −20 °C. Analytes in collected supernatant were measured by ELISA as per manufacturer's instructions (DuoSet ELISA kits, Bio-Techne, Abingdon, UK). Standard curves were fitted to a 4-parameter logistic fit curve in GraphPad Prism v8 software and sample concentrations interpolated from these graphs.

**Quantitation of SAβG activity**. SAβG activity in cultured cells was quantified using the 96-Well Cellular Senescence Activity Assay kit (CBA-231, Cell Biolabs Inc, CA, USA) as described previously[9].

**Statistical analysis and reproducibility**. Chi-square and Fisher's exact test on the decidual pathway defects were performed using online calculators at https://www.socscistatistics.com/. All other analyses were implemented in R. Single-cell RNA-seq was validated on three independent human primary endometrial stromal cell cultures, and six endometrial biopsies in vivo. A seventh sample was excluded from the in vivo analysis due to a considerable batch effect introduced by processing of this sample in isolation from the other six. A minimum of four biological repeats were used for secretome analyses. In addition, 250 timed endometrial biopsies were used to generate the percentile graphs, and 90 and 89 biopsies of control subjects and recurrent pregnancy loss patients, respectively, were analysed. The sample size of clinical samples was determined by the availability of biopsy samples from patients with the appropriate phenotype.

**Reporting summary**. Further information on research design is available in the Nature Research Reporting Summary linked to this article.

## Data availability

The Drop-Seq data were deposited in the GEO repository (http://www.ncbi.nlm.nih.gov/geo/ with accession number: GSE127918). All other data are available on request to the corresponding author.

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

## Acknowledgements

We are grateful to all the women who participated in this research. This work was supported by funds from the Tommy's National Miscarriage Research Centre and Wellcome Trust Investigator Award to J.J.B. and S.O. (212233/Z/18/Z).

## Author contributions

Conceptualization: J.J.B.; methodology: P.V., E.S.L, J.L. and S.O; investigation: E.S.L., P.V., J.M., M.D.C., P.J.B., C-S.K. and K.F.; writing - original draft: J.J.B., P.V. and E.S.L.; funding acquisition: S.Q., S.O. and J.J.B.; resources: J.O., L.J.E., S.Q. and J.J.B.; supervision: S.O. and J.J.B.

## Competing interests

The authors declare no competing non-financial interests but the following competing financial interests: Patent application, The University of Warwick, Professor Jan Brosens, application no. 1911947.8, pending, identification of *SCARA5* and *DIO2* as marker genes

of decidual cells and senescent decidual cells, respectively. The remaining authors declare no competing financial interests.
