## [Peer Review File · Communications Biology]

Reviewers' comments:

Reviewer #1 (Remarks to the Author):

SUMMARY

This manuscript used single cell RNA-seq to investigate the differentiation of primary endometrial stromal cells into decidual cells in vitro. Next, the effect of isolated uterine natural killer (uNK) cells on decidual cells was investigated in vitro. Additionally, scRNA-seq was used to understand the nature of cells in endometrium from the implantation window. Finally, gene expression analyses were used to identify that decidual dyshomeostasis is prevalent in women with a history of recurrent pregnancy loss.

OVERALL AND MAJOR COMMENTS

This manuscript uses a combination of cutting-edge transcriptomics coupled with clinical samples to very nicely test the hypothesis that different subpopulations of decidual cells develop during embryo implantation, respond to uNK cells, and are involved in recurrent pregnancy loss. The results are novel and will influence thinking within the field of pregnancy biology and more widely.

For the most part, the manuscript is well written and contains a series of novel experiments that are appropriately designed, conducted and explained.

Major Comments:

- 1) The data supporting the concept that stromal cells differentiate into acutely senescent and stress-resistant decidual subpopulations is an attractive theory, but is not well supported by mechanistic data. The Introduction should be revised to indicate that IL15-stimulated uNK cells target stressed decidual cells for elimination is a hypothesis rather than a fact. Specific papers supporting that hypothesis should be cited (Line 56). What is the precise data demonstrating that stressed decidual cells are eliminated by uNK cells during early pregnancy?
- 2) A potential problem is that the authors used a filtration method of scRNA-seq data where read count was plotted against UMIs and all cells that fell beyond the inflation point in the distribution curve were not used, resulting in less than 800 cells for each time point. I think that their filtration method is aggressive and may not be rational. To determine cell free RNA (so called 'Ambient RNA'), there is a more reliable method such as those described in the software 'soupX' (<https://rdrr.io/github/constantAmateur/SoupX/man/SoupX.html>) that the authors could compare with their method. The authors need to provide evidence that their filtration method is not too aggressive, and that they have not thrown cells that were authentic for the expression analysis.
- 3) Figure 3: EpC3 should be removed from the aggregate analysis of the endometrial biopsies displayed in Fig. 3A, since it was present in only one endometrial biopsy sample. Supplementary data tables should elaborate on the genes for each of the 5 clusters. More precise information on the number of cells analyzed in the endometrial biopsies and such need to be provided as for the scRNA-seq analysis of the in vitro cultured and differentiated stromal cells.
- 4) The Discussion is very well written, but the Introduction should contain more precise information to support the hypothesis and provide background for the less informed reader.

SPECIFIC COMMENTS

Line Comment

57 hCG also acts on the endometrium with local effects on the glands and stroma. Is it possible that hCG also has differential effects on enSC based on their subpopulation?

75 Please describe the decidualization procedure in more detail for the reader. It would be helpful to have the experimental design placed in Figure 1 if possible.

102 What is the precise criteria for designating specific decidual cell subpopulations as senescent or stress-resistant?

129-131 This sentence is grammatically awkward.

134 The authors should provide supplementary information on genes expressed in the different clusters derived by scRNA-seq on endometrial biopsies.

174 How was gene expression measured?

404 Is the scRNA-seq data from the endometrial biopsies also deposited in GEO?

Reviewer #2 (Remarks to the Author):

In this manuscript, Lucas et al. investigate the uterine mucosa in women with recurrent pregnancy loss (RPL) using single cell RNA sequencing. The focus is on stromal cells and its interactions with uterine Natural Killer (NK) cells and the results show there is a decidualisation defect in women with RPL. There are potentially interesting findings in this work for the reproductive community. However, much of it needs further experiments and validation to robustly determine whether these results are relevant in vivo. Please find below the detailed comments.

Major comments

1. The genes characterising the senescent and stress-resistant cells were defined on the basis on in vitro decidualisation. However, the culture conditions and enzymatic digestion methods to obtain cells for analysis could themselves create a stress/senescence-inducing environment. This is supported by findings that the stress-resistant genes (CRYAB, HSD11B1 and GLRX) are all markers of normal in vivo decidualised stromal cells whilst DIO2 (marker of their senescent decidual cells) is known to be increased in senescent cells grown in vitro from many other systems. This would explain why the two lineages separate and 'branch' in Fig. 1d. Where these two different cell types are located in vivo in the decidua and when they appear are essential to establish that this is not an in vitro artefact and is specific to the decidualisation process in vivo. Can these markers be validated at the protein level in vivo across the cycle?
2. The rationale for introducing NK cells in Fig. 2b and c is not clear. It is stated that the decidual secretome is important but it seems that the co-culture was to discern the effect of uNK cells on stromal cells and not vice versa? There are several issues with this experiment. Please see in detail below under points 3 and 4.
3. In their assays, NK cells have been exposed to IL-15 for 5 days. In order for the readout of these assays to be convincing (Fig., first of all it is necessary to check the purity of isolation and whether they are indeed activated as well as their viability – this is essential as tissue NK cells survive poorly in vitro for more than 24 hours if not activated.
4. How the authors can conclude from their results in Fig. 2c as "selectively killing of acutely senescent decidual cells" (L124) is unclear. Evidence that NK cells are killing stromal cells is completely lacking without performing formal NK cytotoxicity assays. NK cells do not kill normal cells and only virally-infected or transformed cells so this would be an unexpected finding. Why is killing needed if menstruation results in breakdown and elimination of the whole mucosa? Can the authors provide additional evidence and further clarify their statement?
5. There is no description of how NK cells were counted and why number would relate to function (Fig. 5c)? Quantification of NK cells is very difficult as NK numbers change daily during the luteal

phase and these changes also depend on several factors: depth from the surface, amount of edema and gland dilation for example. In L175, it is stated that there are clinically validated tests but there are still as of yet, no evidence based clinical indications for counting NK cells.

6. The results from in vitro system are then compared to stromal cells isolated from implantation stage biopsies. Firstly, the single cell data from biopsies show that NK cells represent ~90% of cells with absence of T cells when there should be albeit at much lower %. The tissue digest did not capture all populations? Furthermore, the presence of monocytes and B cells suggests there is contaminating blood cells present in their tissue biopsy.

Minor comments:

1. The manuscript is full of unusual terms: "impaired fate divergence, immune privileged matrix, branching genes, ripple effects, dyshomeostasis". These kinds of terms are confusing. It would be better to use plain/clear terms. For example, it is stated that implantation creates an 'immune-privileged decidual matrix'. What is this referring to in the human context as the reference refers to murine data?

2. It is possible that artefacts may be introduced due to batch effects. Have any statistical methods been used to correct batch-to-batch effect on the single cell transcriptomic data in vitro and in vivo?

3. Figure 3a. To see the overall contributions of each patient sample to the cell populations, please can you provide a plot showing the cells according to patient origin?

4. The authors show in Fig. 5b the expression of these SCARA5 and DIO2 across the cycle. The methods only state samples from secretory phase, what is the origin of these samples?

Reviewer #3 (Remarks to the Author):

The authors used single-cell RNA sequencing to chart the differentiation process of EnSC towards the decidualization in vitro and in vivo. Afterward, they highlighted that the expression of two genes, SCARA5 and DIO2 have implication in decidual dyshomeostasis. In general, it is appreciated to generate the transcriptome road map of EnSC differentiation. However, it would be better if the authors can further explore the data for novel discovery instead of validating two subpopulations (i.e stress-resistant and senescent) as known before. Followings are my comments:

Major points:

1. For Fig1b, it lacks the necessary explanation of how the 7 transcriptional states are determined. Are these 7 transcriptional states the same to 7 clusters identified by SNN with 10 PCs? How could the authors conclude the split of two EnSC populations is from S4 (Line 89) but not S5? The illustration in Fig1b seems to me that S5 segregated into two clusters that would progress into two states? It would also be more informative to highlight gene expression (Fig1c) back to t-SNE to show when and how these genes are expressed. I am curious why two subpopulations only appears at D8 and how this is regulated.

2. In general, the S5 state is very confusing, which includes cells from D6, D8. Does this state only contain stress-resistant cells? When could senescent cells be detected? As written in the text, it seems to me that resistant and senescent EnSCs appear simultaneously. I also don't understand what is the purpose to do WD stage and what the cell identity of WD cells? Does the withdrawal of signal play any role in senescent/resistant differentiation?

3. For pseudotime analysis, how each 7 cell state is localizing along this trajectory? What cell states of branch 1 and branch 2 correspond to in the Fig1b? In general, Fig1b and 1d are not related or connected. What is the logic behind these two analyses?

4. I don't see the point why the authors do pseudotime analysis per individual (Fig4a), which could

be much noisier due to few cells. I think that using all cells will give a much more robust signal because one can get cells with various developmental states. Then cells can be annotated with individual information to conclude the status of each individual. Also, the authors didn't compare in vitro and in vivo process in a more comprehensive way, such as what is recapitulated in vitro, what is different. More analyses should be done in that direction.

5. I don't think Fig4b is so much informative. It is more reasonable to directly compare the branching genes from in vitro and in vivo single-cell analysis such as by doing Venndigram analysis. Then showing the correlation of those overlapped genes in vitro and in vivo samples.

6. It is not clear to me if SCARA5 and DIO2 are also identified as two subpopulation markers from in vivo samples since the authors only showed the correlation from in vitro study. Also, it is also not clear to me why the authors chose only these two genes among other DEGs in Fig 1e since SCARA5 is not the one that has the most difference. In addition, the authors should also validate other markers to indicate stress-resistant or senescent decidual cell from 250 human samples.

7. I also suggest that the authors do gene regulatory network analysis for those differentially expressed genes they identified in order to understand the possible pathway that drives the resistance and senescence.

8. I am skeptical about using the sum of SCARA5 and uNK centiles as an index to determine the RPL. Such percentage is kind of dynamic depending on the luteal stage, i.e. midluteal or lateluteal and probably other condition of the person. Can the authors really find a panel of genes which show dynamic changes along with implantation time window and show the difference between health and RPL patients? It will be more convincing to rely on only two markers.

Minor points:

1. It would be more informative that the authors provide QC of single cells from in vitro and in vivo experiments in a separate supplementary figure, including mapped reads info and detected genes etc. Also, I didn't find where the authors deposited the data to the public.

2. The authors did three replicates for primary EnSC cultures. But why only one culture has all the timepoints and other two cultures only picked certain timepoints? Also for all cells, t-SNE of FigS3 and Fig.1a are very different. Are the authors using different cells?

3. For co-regulated gene network analysis, I wonder that in the pattern A, A3 is very different from A1 and A2. How this pattern is not identified by itself?

4. EnSCs are collected from mid- and late-luteal endometrium. Can this be reflected in EnSc cluster? Can the authors zoom into EnSc cluster in Fig 3a to see if there are subpopulations of resistant and senescent EnSCs?

We wish to thank the reviewers for their insightful comments and useful suggestions. We have taken on board all comments, added new experimental data, performed additional analyses, and re-written the manuscript for clarity.

Reviewer #1

SUMMARY

This manuscript used single cell RNA-seq to investigate the differentiation of primary endometrial stromal cells into decidual cells *in vitro*. Next, the effect of isolated uterine natural killer (uNK) cells on decidual cells was investigated *in vitro*. Additionally, scRNA-seq was used to understand the nature of cells in endometrium from the implantation window. Finally, gene expression analyses were used to identify that decidual dyshomeostasis is prevalent in women with a history of recurrent pregnancy loss.

OVERALL AND MAJOR COMMENTS

This manuscript uses a combination of cutting-edge transcriptomics coupled with clinical samples to very nicely test the hypothesis that different subpopulations of decidual cells develop during embryo implantation, respond to uNK cells, and are involved in recurrent pregnancy loss. The results are novel and will influence thinking within the field of pregnancy biology and more widely.

For the most part, the manuscript is well written and contains a series of novel experiments that are appropriately designed, conducted and explained.

Authors' response:

We appreciate these supportive comments.

Reviewer #1 (Major Comments):

1) The data supporting the concept that stromal cells differentiate into acutely senescent and stress-resistant decidual subpopulations is an attractive theory, but is not well supported by mechanistic data. The Introduction should be revised to indicate that IL15-stimulated uNK cells target stressed decidual cells for elimination is a hypothesis rather than a fact. Specific papers supporting that hypothesis should be cited (Line 56). What is the precise data demonstrating that stressed decidual cells are eliminated by uNK cells during early pregnancy?

Authors' response:

We thank the reviewer for this comment. As for the inference that our concept is not supported by mechanistic data, we wish to highlight some key findings of our previous papers.

The conclusion that stromal cells differentiate into decidual cells and senescent decidual cells was based on the analysis of a range of senescence markers (e.g. p53, p16^{INK4}, LMNB1, HMGB1, mH2A, H3K9me3) *in vitro* and *in vivo*, the emergence of 'islets' of senescence-associated β -galactosidase (SA β G)-positive cells upon decidualization *in vitro*, the sharp rise in SA β G activity in whole tissue biopsies upon transition of the proliferative to

secretory phase, and quantitative analysis of p16^{INK4}-positive cells in 308 biopsies obtained between LH+6/12 (PMID: 29227245).

We also demonstrated that the initial inflammatory phase upon decidualization of cultured stromal cells coincides with secretion of components of the SASP (senescence-associated secretory phenotype), including mediators of senescent reinforcement such as IL-6 and IL-8. We showed that pre-treatment of undifferentiated stromal cells with dasatinib (a senolytic drug) or palbociclib (a P16^{INK4} mimetic) is sufficient to attenuate or enhance, respectively, the initial secretion of IL-6 and IL-8 and the emergence of SA β G-positive decidual cells. We further reported that various compounds, including the mTOR inhibitor rapamycin (a pharmacological repressor of replicative senescence), SB265610 (a potent CXCR2 inhibitor that blocks paracrine/bystander senescence), and resveratrol (SIRT1 inhibitor), not only block decidualization but also decidual senescence (PMID: 29227245; PMID: 30894514)

Our assertion that uNK cells target and eliminate senescent decidual cells was based on several strands of evidence, including direct visualization of uNK cell-mediated killing *in vitro* by time-lapse microscopy. We provided mechanistic evidence that uNK cell-mediated killing involves perforin- and granzyme-containing granule exocytosis and is inhibited by a blocking antibody against NKG2D (Natural Killer Group 2D), a receptor expressed by all human NK cells and CD8-positive T cells that binds stress ligands expressed on the surface of senescent cells (PMID: 29227245). Further, our observations on uNK-cell mediated killing of senescent decidual cells have been highlighted in two independent review articles on immune surveillance of senescent cells (PMID: 29427795 and PMID: 30811627).

Thus, while we contest the assertion that our work is merely based on an attractive hypothesis, we do acknowledge that the underpinning concepts were not articulated well in the Introduction. This has now been addressed.

The reviewer also questioned the evidence demonstrating that stressed decidual cells are eliminated by uNK cells during early pregnancy. As outlined in the revised manuscript, our data suggest that uNK cell-mediated clearance of senescent decidual cells is of particular importance during the implantation process, although it seems likely that uNK cells continue to play a role in homeostasis of the decidua in early pregnancy. As far as we are aware, targeting of senescent decidual cells by uNK cells in pregnancy has not yet been investigated directly; and unfortunately we do not have access to termination of pregnancy samples. However, it should be noted that the decidua is morphologically grossly abnormal in mice devoid of uNK cells (PMID: 10899912), a phenomenon not yet fully explained.

Reviewer #1

2) A potential problem is that the authors used a filtration method of scRNA-seq data where read count was plotted against UMIs and all cells that fell beyond the inflation point in the distribution curve were not used, resulting in less than 800 cells for each time point. I think that their filtration method is aggressive and may not be rational. To determine cell free RNA (so called 'Ambient RNA'), there is a more reliable method such as those described in the software 'soupX' (<https://rdr.io/github/constantAmateur/SoupX/man/SoupX.html>) that the authors could compare with their method. The authors need to provide evidence that their filtration method is not too aggressive, and that they have not thrown cells that were authentic for the expression analysis.

Authors' response:

The estimate of the true number of cells based on the cumulative read distribution matched the prediction based on bead and cell flow rates and bead inputs into the downstream processing steps. In response to the Reviewer's comment, we have used two additional methods, DropUtils and SoupX R packages. Using DropUtils, we computed the inflection point in the distribution of UMIs as the point on the curve where the first derivative is minimized. Using SoupX, we estimated the fraction of contaminating RNA as a function of the number of UMIs per cell. Both methods agree with our original estimate. We have included two additional supplementary figures to the manuscript containing the DropUtils Inflection plots for the *in vitro* time-course samples (Supplementary Fig. S1) and *in vivo* biopsy samples (Supplementary Fig. S6).

Reviewer #1

3) Figure 3: EpC3 should be removed from the aggregate analysis of the endometrial biopsies displayed in Fig. 3A, since it was present in only one endometrial biopsy sample. Supplementary data tables should elaborate on the genes for each of the 5 clusters. More precise information on the number of cells analyzed in the endometrial biopsies and such need to be provided as for the scRNA-seq analysis of the *in vitro* cultured and differentiated stromal cells.

Authors' response:

Although EpC3 cells were predominantly derived from one sample, they are not exclusively so. Therefore, we believe it is important to retain this population in Figure 3. We now draw the attention of the reader to the origin of this population by marking the population with an asterisk on the figure panel (new Figure 4a) and in the figure legend. We have added two new supplementary tables. Supplementary Table S3 shows the number of cells analysed in each sample and their distribution across cell-types and states. Supplementary Table S6 lists marker genes for each of the 5 main cell-types observed in the biopsy samples.

Reviewer #1

4) The Discussion is very well written, but the Introduction should contain more precise information to support the hypothesis and provide background for the less informed reader.

Authors' response:

We thank the Reviewer and accept this criticism. We have re-written and expanded the Introduction. Briefly, we have added precise definitions for acute and chronic senescence and then applied these to the endometrium. We believe that by doing so, the Introduction both captures our previous observations in human endometrium more clearly and place them in a broader biological context.

Reviewer #1 (specific comments)

57 hCG also acts on the endometrium with local effects on the glands and stroma. Is it possible that hCG also has differential effects on EnSC based on their subpopulation?

Authors' response:

This is an excellent question. As part of a different research project, i.e. the role of LHCG receptor (LHCGR) recycling via early- and very-early endosomes (PMID: 29212031), we

have exhaustively characterised the impact of hCG on primary human endometrial stromal cells. In our hands, we found no consistent effects of hCG on proliferation of stromal cells, induction of decidual marker genes, induction of SA β G activity upon decidualization, cAMP production, or activation of various signalling pathways as determined by a phosphokinase screen. It should be noted that the expression of *LHCGR* in primary stromal cells and in whole endometrial biopsies is extremely low (0-0.3 TPM). However, we did find that hCG consistently increases uNK cell proliferation, as already reported by others (PMID: 19196802; see also our response to Reviewer#2), suggesting a role for hCG signalling (and other embryonic cues) in uNK cell-mediated elimination of senescent decidual cells at implantation.

Reviewer #1

75 Please describe the decidualization procedure in more detail for the reader. It would be helpful to have the experimental design placed in Figure 1 if possible.

Authors' response:

Thank you for these suggestions. The manuscript has been amended accordingly and the experimental design is now depicted in the new Figure 1a. For clarity, we have also added a schematic drawing of the uNK cell co-culture experiments (new Figure 3c)

Reviewer #1

102 What is the precise criteria for designating specific decidual cell subpopulations as senescent or stress-resistant?

Authors' response:

Senescence is context specific and there is no single unequivocal marker of cellular senescence (PMID: 28729727, PMID: 29477613). In the context of the findings in the present study, senescent decidual cells can be described as progesterone-resistant cells that abundantly express extracellular matrix proteins and proteases as well as other SASP components.

Reviewer #1

129-131 This sentence is grammatically awkward.

Authors' response:

The entire paragraph has been re-written for clarity.

Reviewer #1

134 The authors should provide supplementary information on genes expressed in the different clusters derived by scRNA-seq on endometrial biopsies.

Authors' response:

Thank you. The new Supplementary Table S6 shows genes enriched in different populations *in vivo*.

Reviewer #1

174 How was gene expression measured?

Authors' response:

By RT-qPCR. This is now stated explicitly.

Reviewer #1

404 Is the scRNA-seq data from the endometrial biopsies also deposited in GEO?

Authors' response:

Yes. As was stated in the Methods, the Drop-Seq data were deposited in the GEO repository (accession number: GSE127918). We now included a Data Availability statement at end of the Methods section to make it clearer.

Reviewer #2:

In this manuscript, Lucas et al. investigate the uterine mucosa in women with recurrent pregnancy loss (RPL) using single cell RNA sequencing. The focus is on stromal cells and its interactions with uterine Natural Killer (NK) cells and the results show there is a decidualisation defect in women with RPL. There are potentially interesting findings in this work for the reproductive community. However, much of it needs further experiments and validation to robustly determine whether these results are relevant *in vivo*. Please find below the detailed comments.

Authors' response:

For clarity, scRNA-seq analysis of endometrial biopsies was performed to identify marker genes that are specific to differentiating stromal cells *in vivo*, which were then used to examine the decidual responses in clinical biopsies from control subjects and RPL patients. We have taken on board the need for further experiments and validation. Briefly, we now provide:

- kinetic data on the appearance of decidual cells (DC) and senescent decidual cells (snDC) and responses to withdrawal of the differentiation signals (new Fig. 1e),
- new pseudo-time figure (new Fig. 1f),
- additional analysis of co-regulated gene networks and core decidual transcription factors (new Fig. 2),
- additional uNK cell co-culture experiments (new Fig. 3),
- new Figure 5 presenting further analysis of *SCARA5* and *DIO2* expression, including multiplexed single-molecule in situ hybridization analysis of both genes,
- additional bioinformatic analysis of *in vivo* endometrial stromal cells expressing high *SCARA5* but low *DIO2* mRNA levels versus cells with low *SCARA5* but high *DIO2* mRNA levels (new Supplementary Fig. S9).

Reviewer #2

1. The genes characterising the senescent and stress-resistant cells were defined on the basis on *in vitro* decidualisation. However, the culture conditions and enzymatic digestion methods to obtain cells for analysis could themselves create a stress/senescence-inducing environment. This is supported by findings that the stress-resistant genes (*CRYAB*, *HSD11B1* and *GLRX*) are all markers of normal *in vivo* decidualised stromal cells whilst *DIO2* (marker of their senescent decidual cells) is known to be increased in senescent cells

grown *in vitro* from many other systems. This would explain why the two lineages separate and 'branch' in Fig. 1d. Where these two different cell types are located *in vivo* in the decidua and when they appear are essential to establish that this is not an *in vitro* artefact and is specific to the decidualisation process *in vivo*. Can these markers be validated at the protein level *in vivo* across the cycle?

Authors' response:

As reported previously (PMID: 29227245, Figure 1—figure supplement 1), even after 6 passages (i.e. 60 days in continuous culture), exposure of endometrial stromal cells to a decidualogenic stimulus enhances senescence-associated β -galactosidase (SA β G) activity and triggers the appearance of SA β G-positive islets, which strongly argues against the suggestion that the emergence of decidual subsets *in vitro* is a consequence of enzymatic digestion of the tissue. The emergence of senescent cells upon decidualization has also been documented by a Japanese team (PMID: 30894514)

As outlined in our response to Reviewer #1, we have taken several approaches to demonstrate the emergence and accumulation of senescence decidual cells during the luteal phase of the cycle *in vivo*, including measurement of SA β G activity in 73 biopsies obtained across the cycle, quantitative analysis of p16^{INK4}-positive stromal and epithelial (glandular and luminal) cells in 308 LH-timed biopsies, and Western blot analysis on protein lysates from whole tissue biopsies for p53, p16^{INK4}, LMNB1, HMGB2, H3K9me3 and mH2A levels. Note that commonly used senescence markers, including SA β G, p53, p16^{INK4} and histone modifications, are regulated post-transcriptionally. Nuclear accumulation of p53 without corresponding increase in mRNA expression in primary decidualizing cultures and during the late-secretory phase of the cycle was first documented 15 years ago (PMID: 15472230).

In response to the Reviewer's request for evidence that *SCARA5* and *DIO2* mRNA expression mark different stromal cells *in vivo*, we performed multiplexed, single-molecule *in situ* hybridization for both transcripts on endometrial biopsies. The data are presented in the new Figure 5d. The images show that *DIO2* and *SCARA5* transcripts mark distinct cells *in vivo* as well as cells in an intermediate state.

Reviewer #2

2. The rationale for introducing NK cells in Fig. 2b and c is not clear. It is stated that the decidual secretome is important but it seems that the co-culture was to discern the effect of uNK cells on stromal cells and not vice versa? There are several issues with this experiment. Please see in detail below under points 3 and 4.

Authors' response:

We apologise for the lack of clarity. We hope that the revised Introduction now makes the rationale for these experiments obvious. The co-culture experiments were indeed designed to examine if uNK cells selectively target senescent decidual cells responsible for the SASP. Indeed, we did not examine the impact on uNK cells but hope to address this in future experiments. Please note that additional experiments are now shown in the new Figure 3.

Reviewer #2

3. In their assays, NK cells have been exposed to IL-15 for 5 days. In order for the readout of these assays to be convincing (Fig., first of all it is necessary to check the purity of isolation and whether they are indeed activated as well as their viability – this is essential as tissue NK cells survive poorly *in vitro* for more than 24 hours if not activated).

Authors’ response:

The method of uNK cell isolation was described in detail in our previous publication (PMID: 29227245). Briefly, uNK cells are isolated from the supernatant of freshly digested stromal cell cultures 16-18 hours post-seeding by magnetic activated cell sorting (MACS) using PE-conjugated CD56 (NCAM1) antibody. This approach yields highly purified uNK cells (see Figure below; PMID: 29227245), which continue to proliferate in culture. As stated in response to Reviewer 1, uNK cell proliferation *in vitro* is enhanced by hCG (Figure 1b below), as reported previously (PMID: 19196802). Targeting of decidual cells by uNK cells was also captured in co-culture by time-lapse microscopy (Figure 1c below and PMID: 29227245), which in turn attests to the viability of the cells. Further, the new Figure 3d show SA β G activity in undifferentiated and decidualized stromal cells co-cultured or not with uNK cells.

Figure 1 (for review purpose only). **a**, CD56 labelling of MACS-isolated uNK cells after cytopsin preparation. **b**, XTT proliferation assay demonstrating uNK proliferation in culture, which is enhanced in response to hCG stimulation. **c**, images from time-lapse microscopy

demonstrating targeting and killing of a selected decidual cells by uNK cells in co-culture (from PMID: 29227245). The time of recording (hours: min) is shown in the right bottom corner of the images.

Reviewer #2

4. How the authors can conclude from their results in Fig. 2c as “selectively killing of acutely senescent decidual cells” (L124) is unclear. Evidence that NK cells are killing stromal cells is completely lacking without performing formal NK cytotoxicity assays. NK cells do not kill normal cells and only virally-infected or transformed cells so this would be an unexpected finding.

Authors’ response:

We wish to point out that multiple studies have documented selective killing of senescent cells by NK cells in different tissues (e.g. PMID: 18724938, PMID: 22751116, PMID: 24800169, PMID: 17251933, PMID: 31160572). As aforementioned, our observations on uNK-cell mediated killing of senescent decidual cells have been highlighted in two recent, independent review articles on immune surveillance of senescent cells (PMID: 29427795 and PMID: 30811627).

The experimental approaches we employed in our previous publication are, for reviewing purposes only, depicted below.

Briefly, we reported (PMID: 29227245):

- uNK killing activity in co-cultures captured in real-time using time-lapse microscopy,
- loss of cellular confluency in co-cultures using real-time using xCELLigence proliferation assay,
- loss of killing activity in co-cultures when incubated with an IL-15 blocking antibody,
- loss of killing activity in co-cultures incubated with a blocking antibody against major NK cell-surface recognition receptor (NKG2D),
- loss of killing activity in co-cultures incubated with a granzyme B inhibitor (3,4-DCI),
- loss of killing activity in co-cultures incubated with the pan-caspase inhibitor Z-VAD-FMK,
- no evidence of uNK cell-mediated killing of undifferentiated stromal cells.

Reviewer #2 Why is killing needed if menstruation results in breakdown and elimination of the whole mucosa? Can the authors provide additional evidence and further clarify their statement?

Authors' response:

Menstrual shedding of the superficial layer is obviously a very effective way of eliminating senescent endometrial cells. Indeed, menstruation could arguably be viewed as the most conspicuous example in human biology of self-elimination of senescent cells. Instead, we argue that, in a conception cycle, uNK cell-mediated killing of senescent decidual cells is essential to avoid emergence of chronic senescence, breakdown of the decidual-placental interface, and miscarriage.

We appreciate the criticism that we failed to provide sufficient context for our observations. Briefly, senescent cells are resistant to apoptosis, metabolically active and produce large amounts of soluble factors collectively called senescence-associated secretory phenotype (SASP). Two main classes of senescent cells have been identified in many tissues and organs: acute and chronic senescent cells (PMID: 28029153). Acute senescent cells are generated during coordinated, beneficial biological processes characterized by a defined senescence trigger, transient senescent-cell signalling functions, and eventual senescent-cell clearance. In contrast, chronic senescence and sustained SASP-associated inflammatory signalling leads to senescence in neighbouring cells (a phenomenon referred to secondary or bystander senescence), ECM remodelling, immune cell infiltration and loss of tissue function.

These features are recapitulated upon decidualization of primary endometrial stromal cells: the process starts with an acute inflammatory response, which – after a lag-period of 4 days -leads to decidual cells (S5) as well as the emergence of a limited number of senescent decidual cells (S6). Without co-culturing uNK cells, the number of senescent decidual cells increases rapidly in parallel with a decrease in decidual cells, indicative of secondary/bystander senescence. The manuscript has been revised to ensure that these concepts are more clearly articulated.

In cycling endometrium, the levels of p16^{INK4}-positive stromal cells first peak during the mid-luteal phase (i.e. coinciding with the implantation window), then decline transiently (in concert with expansion of uNK cells) before rising sharply again prior to menstruation (presumably reflecting falling P4 levels). The data presented here indicate that RPL is associated with excessive accumulation of (chronic) senescent cells [reflecting either stem cell deficiency (PMID: 26418742), uNK cell deficiency or dysfunction, or both] and lack of decidual cells. Our prediction is that this decidual state at implantation will inevitably lead to a dysfunctional placental-maternal interface prone to breakdown and bleeding.

We wish to emphasize that this pathological pathway is not merely of academic interest but may lead to new strategies for miscarriage prevention. For example, we recently completed the SIMPLANT study, a double-blind, randomised, placebo-controlled pilot trial testing if sitagliptin (DPP4 inhibitor) given over 3 cycles increase the abundance of endometrial mesenchymal stem-like progenitor cells (measured by colony forming unit assays) in RPL patients (EU Clinical Trials Register no. 2016-001120-54). While we hope to publish soon, we can disclose that sitagliptin over 3 cycles increases the abundance of clonal cells by

average by 68% and decreases *DIO2* mRNA levels by 40%. A larger follow-up trial is planned to examine if pre-conception sitagliptin improves live birth rates in RPL patients

Reviewer #2

5. There is no description of how NK cells were counted and why number would relate to function (Fig. 5c)? Quantification of NK cells is very difficult as NK numbers change daily during the luteal phase and these changes also depend on several factors: depth from the surface, amount of edema and gland dilation for example. In L175, it is stated that there are clinically validated tests but there are still as of yet, no evidence based clinical indications for counting NK cells.

Authors' response:

We apologise for this oversight and the Methods have been amended accordingly. The test used has been standardized across different laboratories in the UK (PMID: 27214130) and is based on quantification of CD56-positive uNK cells relative to CD56-negative stromal cells in 3 high-power fields underlying the luminal epithelium, thus negating the concerns regarding the depth, presence of edema, and degree of glandular dilatation.

The Reviewer is right in stating that the abundance of uNK cell does not yield insights into the function of these cells in cycling endometrium. However, the abundance of uNK cells during the luteal phase varies dramatically between patients (from <1% to >50% of stromal cells) even when adjusted for LH day (centile graphs) and between cycles (in keeping with their role in tissue homeostasis). The work presented here indicates that the pathological significance of high or low uNK cells depends on the state of the stroma, i.e. the abundance of senescent decidual cells and decidual cells. The whole purpose of the present study was to identify biomarkers that can potentially be used to measure decidual subsets in clinical samples, thus providing the essential context to interpret uNK cell measurements. We envisage that this approach can be refined further, for example by including KIR genotyping, and may lead to the development of a preconception screening test to identify women at increased risk of (euploid) miscarriage.

Reviewer #2

6. The results from in vitro system are then compared to stromal cells isolated from implantation stage biopsies. Firstly, the single cell data from biopsies show that NK cells represent ~90% of cells with absence of T cells when there should be albeit at much lower %. The tissue digest did not capture all populations? Furthermore, the presence of monocytes and B cells suggests there is contaminating blood cells present in their tissue biopsy.

Authors' response:

We thank the Reviewer for highlighting potential biases in the *in vivo* RNA-seq data. We are indeed aware of a number of biases. For example, endometrial glands are much more resistant to digestion than stromal cells, which means that epithelial cells are relatively underrepresented in the scRNA-seq data. Significant blood contamination is less likely. First, most Pipelle biopsies are free of macroscopic blood contamination (i.e. no visible discolouration when placed in clear liquid). Second, T cell-rich lymphoid aggregates, which also contain macrophages and B cells, are indeed present in secretory endometrium but localize predominantly in the basal layer (PMID: 9103229), which is not sampled by a Pipelle

biopsy. B cells and macrophages are also present in the superficial layer. Finally, significant contamination with blood immune cells would arguably lead to dilution of the uNK cell fraction.

Reviewer #2

Minor comments:

1. The manuscript is full of unusual terms: “impaired fate divergence, immune privileged matrix, branching genes, ripple effects, dyshomeostasis”. These kinds of terms are confusing. It would be better to use plain/clear terms. For example, it is stated that implantation creates an ‘immune-privileged decidual matrix’. What is this referring to in the human context as the reference refers to murine data?

Authors’ response:

Thank you, we acknowledge this important point. Unusual terms and jargon can distract from the core research findings. Consequently, we have thoroughly re-written the manuscript for clarity, including a changed title.

Reviewer #2

2. It is possible that artefacts may be introduced due to batch effects. Have any statistical methods been used to correct batch-to-batch effect on the single cell transcriptomic data in vitro and in vivo?

Authors’ response:

To reduce the chance of experimental batch-effects as far as possible, library amplification, tagmentation steps and sequencing were performed together within the time-course sets (two batches: Fig1 time-course as a batch, supplementary time-courses as a batch) and in vivo sample set (P1 alone, P2-P7 as a batch). In the case of the biopsy data, while samples P2-P7 were batch processed, sample P1 was run independently and sequenced to a greater depth. To minimize the effect of the sequencing depth on the single-cell analysis, we down-sampled the raw counts for P1 to make the data comparable to the rest of the biopsy samples. However, we acknowledge that sample P1 was different from the other samples and, hence, we have excluded this sample in the revised manuscript. As part of our analysis pipeline, we examined the distribution of nUMI, nGene and percent mitochondria per sample, and scaled the data on nUMI and percent mitochondria. We now include the nUMI, nGene and percent mitochondria distributions per sample for the time-course data in Supplementary Fig. S1 and for the *in vivo* data in Supplementary Fig. S6.

Reviewer #2

3. Figure 3a. To see the overall contributions of each patient sample to the cell populations, please can you provide a plot showing the cells according to patient origin?

Authors’ response

The *t*-SNE plot of *in vivo* data is now presented in the new Figure 4a (upper panel) annotated to indicate the LH-day of the biopsy. In addition, the new Supplementary Table S3 lists the contribution of individual samples to the various cell populations.

Reviewer #2

4. The authors show in Fig. 5b the expression of these SCARA5 and DIO2 across the cycle. The methods only state samples from secretory phase, what is the origin of these samples?

Authors' response:

We regret this oversight. The data were retrieved from GEO (GDS2052). This is now stated explicitly in the body of the text and figure legend.

Reviewer #3 (Remarks to the Author):

The authors used single-cell RNA sequencing to chart the differentiation process of EnSC towards the decidualization in vitro and in vivo. Afterward, they highlighted that the expression of two genes, SCARA5 and DIO2 have implication in decidual dyshomeostasis. In general, it is appreciated to generate the transcriptome road map of EnSC differentiation. However, it would be better if the authors can further explore the data for novel discovery instead of validating two subpopulations (i.e. stress-resistant and senescent) as known before. Followings are my comments:

Authors' response:

The data generated in this study represent indeed an important resource. We have added additional analysis of co-regulated gene networks and core decidual transcription factors (new Fig. 2), which in turn provide additional insights into the mechanisms that drive divergence of decidual cells and senescent decidual cells. There is also a clinical imperative to identify endometrial biomarkers associated with recurrent pregnancy loss. We hope that the revised manuscript provides a better balance between novel observations and clinical relevance.

Reviewer #3 (Major points):

1. For Fig1b, it lacks the necessary explanation of how the 7 transcriptional states are determined. Are these 7 transcriptional states the same to 7 clusters identified by SNN with 10 PCs? How could the authors conclude the split of two EnSC populations is from S4 (Line 89) but not S5? The illustration in Fig1b seems to me that S5 segregated into two clusters that would progress into two states? It would also be more informative to highlight gene expression (Fig1c) back to t-SNE to show when and how these genes are expressed. I am curious why two subpopulations only appears at D8 and how this is regulated.

Authors' response:

Yes, the 7 transcriptional states refer to the 7 clusters identified by SNN with 10 PCs. We do apologise for giving the wrong impression that decidualizing stromal cells instantaneously diverge into two non-dynamic subsets: S5 (decidual cells, DC) and S6 (senescent decidual cells, snDC) after day 4 of decidualization. This was misleading and thank the Reviewer for highlighting this issue. As shown in the new Figure 1e, on day 6 the proportion of cells in the DC and snDC subsets was 78% and 13%, respectively. By day 8, this had changed to 41% and 45%, respectively. This 'switch' in cell state over time reflects secondary/bystander senescence and highlights the need for uNK cell-mediated killing of senescent decidual cells.

Reviewer #3

2. In general, the S5 state is very confusing, which includes cells from D6, D8. Does this state only contain stress-resistant cells? When could senescent cells be detected? As written in the text, it seems to me that resistant and senescent EnSCs appear simultaneously. I also don't understand what is the purpose to do WD stage and what the cell identity of WD cells? Does the withdrawal of signal play any role in senescent/resistant differentiation?

Authors' response:

As outlined above, we acknowledge that the description was incorrect and we have rectified this. The withdrawal experiment demonstrates that DC (S5) de-differentiate rapidly whereas this response is much less pronounced in snDC (S6), suggesting loss of progesterone / cAMP-dependency. Further evidence for progesterone resistance in snDC is now provided in the new Figure 2, showing that snDC express transcription factors known to be repressed by the activated progesterone receptor (PGR) (PMID: 18511503). In fact, *DIO2* is also a PGR-repressed gene.

Reviewer #3

3. For pseudotime analysis, how each 7 cell state is localizing along this trajectory? What cell states of branch 1 and branch 2 correspond to in the Fig1b? In general, Fig1b and 1d are not related or connected. What is the logic behind these two analyses?

Authors' response:

While t-SNE analysis imposes a discrete framework, pseudotime measures changes in cell state as a function of progress along a continuous trajectory. As such, both analyses are complementary. For clarity, we have generated a new pseudotime figure (Figure 1f) that is restricted to day 2 to day 8 (i.e. confined to cells under continuous 8-bromo-cAMP and MPA stimulation) and overlaid cell states. This analysis revealed that decidualizing stromal cells progress along a continuous trajectory towards senescence (scDC). The trajectory was interrupted by a single branchpoint, marking the divergence of DC and a subset of DC already transitioning towards a senescent phenotype. As outlined in the Discussion, our findings suggest that there is a narrow window during the implantation process during which uNK cell-mediated clearance of snDC allows divergence from the default decidual trajectory towards senescence.

Reviewer #3

4. I don't see the point why the authors do pseudotime analysis per individual (Fig4a), which could be much noisier due to few cells. I think that using all cells will give a much more robust signal because one can get cells with various developmental states. Then cells can be annotated with individual information to conclude the status of each individual. Also, the authors didn't compare in vitro and in vivo process in a more comprehensive way, such as what is recapitulated in vitro, what is different. More analyses should be done in that direction.

Authors' response:

The pseudotime analysis in the original Figure 4a was performed on stromal cells from all 7 biopsies and then overlaid with cells from each sample. The purpose was to illustrate a time-dependent transcriptome changes in stromal cells *in vivo*. We have deleted this figure in the revision as meaningful mapping and analysis of branching points *in vivo* requires more

samples across more days in the cycle. We generated a new t-SNE plot annotated to show LH+8 and LH+10 samples (new Fig. 4a). We show that progression from LH+8 to LH+10 is associated with differential expression of 518 genes in stromal cells (new Supplementary Table S7), 49% of which are also part of the 7 co-regulated gene networks (new Figure 3) underpinning the decidual pathway *in vitro*

Reviewer #3

5. I don't think Fig4b is so much informative. It is more reasonable to directly compare the branching genes from *in vitro* and *in vivo* single-cell analysis such as by doing Venndigram analysis. Then showing the correlation of those overlapped genes *in vitro* and *in vivo* samples.

Authors' response:

We believe that this analysis is informative as it shows that the dynamics of branching genes identified *in vitro* is, at least partly, maintained *in vivo*. In response to the Reviewer's comment, we have placed this figure in supplementary data (new Supplementary Figure S8)

Reviewer #3

6. It is not clear to me if SCARA5 and DIO2 are also identified as two subpopulation markers from *in vivo* samples since the authors only showed the correlation from *in vitro* study. Also, it is also not clear to me why the authors chose only these two genes among other DEGs in Fig 1e since SCARA5 is not the one that has the most difference. In addition, the authors should also validate other markers to indicate stress-resistant or senescent decidual cell from 250 human samples.

Authors' response:

The focus in the manuscript was on the top 50 branchpoint genes identified *in vitro*. To be meaningful a marker gene should be highly enriched in stromal cells, not regulated in glandular epithelium, and have a temporal profile across the luteal phase commensurate with the expected switch of DC to snDC prior to menstruation. Out of 50 branch genes, 5 genes (*TIMP3*, *IGF2*, *DIO2*, *SCARA5* and *ABI3BP*) were highly enriched in stromal cells compared to epithelial, endothelial or immune cells. Next, we cross-referenced all 5 genes against RNA-seq data of laser-captured endometrial glands on LH+5, LH+8, and LH+11 (GEO ID: GSE84169 ; PMID: 28974690) to exclude significant time-dependent regulation in the endometrial epithelial compartment as this would potentially introduce a major bias in clinical sample analysis. For reviewing purposes, the data are presented below.

We further cross-referenced the genes against microarray data of endometrial biopsies obtained across the cycle (GEO ID: GDS2052). Only *SCARA5* and *DIO2* met all the criteria.

Importantly, we have now included several additional strands of data supporting our conjecture that *SCARA5* and *DIO2* are meaningful marker genes, including:

- new Figure 5a shows that *DIO2* and *SCARA5* are part of different co-regulated gene networks underpinning snDC and DC, respectively. Note that expression of *DIO2* is a putative marker of progesterone-resistance (PMID: 18511503),
- new Figure 5d: multiplexed single-molecule in situ hybridization analysis of both genes in endometrial biopsies,
- differential gene expression and gene ontology analysis of *in vivo* endometrial stromal cells expressing high *SCARA5* but low *DIO2* mRNA levels versus cells with low *SCARA5* but high *DIO2* mRNA levels (new Supplementary Fig. S9).

Reviewer #3

7. I also suggest that the authors do gene regulatory network analysis for those differentially expressed genes they identified in order to understand the possible pathway that drives the resistance and senescence.

Authors' response:

Thank you for this suggestion. The new Figure 2 now shows the 7 networks of co-regulated genes annotated with core decidual transcription factors. We further use network analysis in the new Figure 3a and Figure 5a.

Reviewer #3

8. I am skeptical about using the sum of *SCARA5* and uNK centiles as an index to determine the RPL. Such percentage is kind of dynamic depending on the luteal stage, i.e. midluteal or lateluteal and probably other condition of the person. Can the authors really find a panel of genes which show dynamic changes along with implantation time window and show the difference between health and RPL patients? It will be more convincing to rely on only two markers.

Authors' response:

We do not propose to use the sum SCARA5 and uNK centiles as an index to determine the RPL. The sole purpose was to illustrate that this simple approach has the potential to stratify RPL patients and control subjects. We are also confident that additional marker genes will be identified upon reconstruction of the entire luteal phase in control and RPL patients using single-cell analysis. At present, this is aspirational and beyond the scope of the current investigation.

Reviewer #3 (Minor points)

1. It would be more informative that the authors provide QC of single cells from in vitro and in vivo experiments in a separate supplementary figure, including mapped reads info and detected genes etc. Also, I didn't find where the authors deposited the data to the public.

Authors' response:

We are including this information in the new Supplementary Figures S1 and S6.

Reviewer #3

2. The authors did three replicates for primary EnSC cultures. But why only one culture has all the timepoints and other two cultures only picked certain timepoints? Also for all cells, t-SNE of FigS3 and Fig.1a are very different. Are the authors using different cells?

Authors' response:

Replicates were performed for the most informative timepoints. Figure S3 differs from Figure 1b as it incorporates data from replicate time-courses on 3 different primary cultures.

Reviewer #3

3. For co-regulated gene network analysis, I wonder that in the pattern A, A3 is very different from A1 and A2. How this pattern is not identified by itself?

Authors' response:

The networks are now described in detail in the Results (L168-198). Please note that the co-regulated pattern A3 has been relabelled as pattern D.

Reviewer #3

4. EnSCs are collected from mid- and late-luteal endometrium. Can this be reflected in EnSc cluster? Can the authors zoom into EnSc cluster in Fig 3a to see if there are subpopulations of resistant and senescent EnSCs?

Authors' response:

t-SNE analysis on stromal cells from individual biopsies invariably show two subsets. This was no longer apparent when the samples are pooled. Further, DEG between these subsets vary substantially between samples and therefore it was not possible to draw robust conclusions. In the absence of single-cell data on many more biopsies across the luteal base, our approach had to be based on in-depth analysis of putative marker genes.

Reviewers' comments:

Reviewer #1 (Remarks to the Author):

This manuscript uses a combination of cutting-edge transcriptomics coupled with clinical samples to very nicely test the hypothesis that different subpopulations of decidual cells develop during embryo implantation, respond to uNK cells, and are involved in recurrent pregnancy loss. The results are novel and will influence thinking within the field of pregnancy biology and more widely.

Overall, the authors have done a very good job addressing the comments from the initial review. This reviewer is very satisfied with the author response to the initial review.

Reviewer #2 (Remarks to the Author):

Generally, the authors have responded to some of the points raised. However, the critical experiments have not been tackled (see below for details). Therefore, the robustness of the data, its relevance to the situation in vivo and the authors' interpretation of the data, are still questionable. Specifically, the issues focus on three main points:

1. The characterisation of the decidual stromal subpopulations emerging during decidualisation in vitro

According to new Supp. Fig.S4, only cultures from Patient 'a' were sequenced for the entire decidualisation time course. The timepoints during which the different subpopulations begin to emerge, D4, D6 and WD, cells only from patient 'a' are shown. Why have cells from Patients 'b' and 'c' been removed from these crucial timepoints? It is fundamental to have biological replicates as evidence that these populations reproducibly emerge during the in vitro decidualisation experiments, especially if one of the points of this paper is to describe these novel populations. It is not possible to say that this pattern and these transcriptional states are not just specific to patient 'a' or that particular decidualisation experiment?

The 'senescent' cells identified have many of the phenotypic hallmarks of differentiated endometrial stromal cells. These include IL-15, ST2, clusterin, CRYAB, HSD11B1, GLRX, GLUL and ALDH1A1. Differentiated stromal cells also stop proliferating and it is not clear why these 'senescent' cells are not just at the terminal end of the differentiation pathway?

The in vivo validation of the scRNA-seq data for the two decidual stromal subpopulations is performed by ISH for DIO2 and SCARA5 in new Fig. 5d. This figure is unclear. Where are these stromal populations with respect to their localisation within the endometrium? Are they all showing corresponding areas of the endometrium and is the staining pattern for each patient consistent across the entire tissue? The interpretation of these results should be assisted by staining for additional canonical senescent markers such as CDKN2A, on a serial section from the same tissue sample.

2. Comparison of stromal populations in women with RPL

The differences between biopsies from women with RPL and controls is of potential interest. It would then be expected that if the response of the stromal cells to progesterone is defective then IL-15 will also be reduced resulting in fewer NK cells. This may just reflect an overall delayed or altered response to progesterone. How does this data fit with other reports (some from this group) claiming uNK numbers are increased in women with RPL (eg. Quenby et al., Fertil Steril 2005; Tuckerman et al., J Reprod Immunol 2010)?

3. Selective killing of senescent stromal cells by uNK cells

There may indeed be a higher proportion of 'stressed' stromal cells in some women with RPL but to extend this as abnormal chronic senescent cells requiring elimination by uNK cells is an unlikely scenario. uNK cells do not normally kill 'self' cells. The paper that the authors' cite in the rebuttal are their own (Brighton et al., eLIFE 2017). In this manuscript, as in Brighton et al., the protocol used for the isolation of uNK cells isolation is problematic. Their protocol will alter uNK phenotype and killing capacity and in vitro experiments will therefore not reflect the situation in vivo. Purity, viability and activation status of the uNK cells must be assessed before their functional experiments are performed to ensure that the results from their co-culture experiments are not due to in vitro artefacts. This must be done by flow cytometry as these characteristics cannot be properly assessed by CD56 on cytopins, shown in their rebuttal (this data is also not in their previous publication).

In the author's rebuttal, they cite several papers to support their claim uNK cells can kill 'senescent' cells. However some of these papers (PMID: 24800169, PMID: 17251933) are in the context of another tissue and a pathological situation of tumours, in which it is known that NK cells do kill.

Their experimental set-up for Fig. 3d was used in their previous paper (Brighton et al., eLIFE 2017, Fig. 5) and it does not show whether only the senescent cells are being killed. All this assay shows is that there is less SA- β -gal produced and there could be several explanations for this. To demonstrate that uNK cells are able to kill 'self' cells that are 'senescent', they need to separate the 'senescent' cells from the other stromal cells and perform conventional NK killing assays (after assessing uNK status as requested above).

If it proposed that the NK receptor mediating this is NKG2D which binds to stress ligands on stromal cells, this needs demonstrating. ULBPs and MICA/B do not appear in their gene list in Fig 1g. Furthermore, NKG2D is not expressed on all uNK cells (Marlin et al., PLOS One 2012).

Additional comments:

The rationale of Fig 4a is not clear in relation to the rest of the paper. The in vivo scRNA-seq profiling of the endometrium still does not show all the immune populations known to be present in the endometrium. NK cells and macrophages are present in the functionalis layer as well as T cells. But there are no B cells in the functionalis. Why do they only find B cells and no T cells in their samples and why are there monocytes? These are only present in blood.

Please clarify the following:

- Introduction p3L62 – 'hyperinflammation due to placentation, under physiological conditions.' What does 'hyper' mean?
- Results p7L163 – 'decidualization is a multistep process that starts with acute auto-inflammatory response.' The use of the term 'inflammatory' is used in this manuscript in relation to placentation and in the initial phases of the decidualisation process. It is not clear why this term is used as inflammation cannot be simply defined by the presence of pleiotropic cytokines such as IL-6. Inflammation is defined by an influx of inflammatory cells (neutrophils, macrophages and other immune cells), deposition of fibrin and edema followed by tissue repair. These are not seen in the early secretory phase (although edema alone is seen in the mid-secretory phase) or the early decidua. The influx of neutrophils occurs at menstruation.

Reviewer #3 (Remarks to the Author):

In general, I think that the authors did a good job of revising the paper. They have added new

analyses to address my concerns including comparing to the previous dataset.
I have no more comments on the paper and support its publication.

Reviewer #4

Having read the manuscript, reviewer 2 comments and the authors' rebuttal, I believe that the manuscript by Lucas et al. provides new insights regarding the molecular pathways of human decidualization. The single cell RNAseq analysis performed on in-vitro cultured decidualizing cells is very informative. The identification of specific markers of divergent decidualization response and findings that these are differentially expressed in endometrial biopsies of RPL patients compared to controls are important. However, Reviewer 2 has raised a couple of important points with regards to the main claims of the paper and data interpretation that I tend to agree with as detailed below.

With regards to point 3 raised by the reviewer, I am in agreement with the reviewer that it is unlikely that NK cells eliminate "senescent" decidual cells in-vivo. NK cells do not have "killing" properties against decidual cells and are actually thought to be important in mediating immune tolerance to the fetus. The evidence presented to that effect of eliminating senescent cells by the authors is based on in-vitro studies in both the current work as well as their related referenced paper (Brighton et al., eLIFE). On both occasions, NK cells are shown to be cytotoxic towards decidual cells compared to non-decidualized cells in general. To support their claim, and as suggested by reviewer 2, it would be essential to separate senescent cells from normal decidual cells and show a selective cytotoxic effect against the senescent decidual cells. Moreover, in-vivo evidence to support such claim is seriously lacking, and currently available in-vivo evidence in mice is not supportive of their claim. Alymphoid mice (lacking NK cells) and IL-15 KO mice that also lack decidual NK cells have alteration in spiral artery remodeling but otherwise have normal fertility, decidualization, normal pregnancies and litter sizes (PMID 22187674, PMID 24000237). Thus, the argument that NK cells are required for elimination of senescent cells for pregnancy success is very speculative. In-vivo the uterine microenvironment has many other immune and other cell types that mediate immunomodulation which is likely very different than the in-vitro system. What is needed is to show in-vivo evidence. For example, electron microscopy showing the proximity between NK cells and the senescent cells in human endometrial tissue; A mouse model showing the effects of NK cell enhancement or inhibition with NK cell antibody and showing whether this leads to elimination or accumulation of the decidual senescent cell phenotype, respectively, would help strengthen such claims.

With regards to the reviewer's additional comment on the immune populations in the scRNAseq analysis, the presence of a small subset of B-cells and some monocytes is likely due to some contamination from circulating leukocytes, as eluded to by the authors' answer in the rebuttal. As the authors rightfully acknowledge, the small number of immune cells analyzed (352 cells only) limits the depth of the analysis. Overall, the large abundance of NK cells is expected. In Figure 4, the population identified as 'monocytes' (IC2) actually has strong expression of CD4, a T cell marker, as well as expression of several dendritic cell markers. This suggests to me that IC2 population may be misidentified and likely contains the T cells that the reviewer found to be missing. This can probably be addressed by re-analysis of that cluster to look for typical T cell markers.

With regards to point 1 of the reviewer, I agree with the reviewer that in vivo validation of the scRNA-seq data would be greatly enhanced by addition of canonical senescent markers such as CDKN2A, to the current validation which is based on ISH for DIO2 and SCARA5.

With regards to point 2 of the reviewer, the literature regarding NK cells abundance in the endometrium of women with RPL is conflicting, and the authors' response is satisfactory in my mind.

Reviewer #5

Point 1 (Supp FigS4). The reviewer asks about data from other patients to make sure data are reproducible. Author says data are "representative of responsive primary cultures". I would agree with the Reviewer here that results from other patients would make the data stronger.

Point 2. About senescent vs terminally differentiated cells. I think the Author's response here is reasonable.

Point 3. in vivo validation of the scRNA-seq data. The Author's response seems adequate here.

Point 4. Comparison of stromal populations in women with RPL. This response seems reasonable too and the associated data (for information only) are convincing.

Point 5. Selective killing of senescent stromal cells by uNK cells. I think the Author's response about cell preparation is fine, however the Reviewer's request to show FACS data of uNK cells is also reasonable and ought to be included.

Point 6. Claim that uNK cells can kill 'senescent' cells. I'm afraid I know of no published data showing this directly. The papers mentioned by the Author are indeed about non-malignant conditions, however not from the uterus, so I am not sure they really help the point of the Author, since every tissue-resident cell will have a tissue-specific biology.

Point 7. Their experimental set-up for Fig. 3d was used in their previous paper (Brighton et al., eLIFE 2017, Fig. 5) and it does not show whether only the senescent cells are being killed. The Author's response seems reasonable here.

Point 8. NKG2D and its ligands. The Reviewer is right here that there is no data supporting the involvement of this receptor-ligand and the authors may want to show these data or not mention this R-L interaction at all, or make sure it is only a speculation.

In conclusion, and having read the interesting paper, my suggestion is to accept it but perhaps change the title (and the relevant text in the paper) to a more cautious:

"Recurrent pregnancy loss may be associated with a pro-senescent decidual response during the peri-implantation window". Rather than is associated with...

Also, I have noticed discrepancies between the Vento-Tormo paper (Nature 2018) and the RNAseq data published here from the three subsets of dNK cells. Specifically, in the Vento-Tormo paper EPAS1, GZMA and GNLY are highly expressed in dNK1, whereas in this paper they are highly expressed in dNK2. Perhaps the Authors ought to comment on this, rather than misleadingly pointing at the similarities only.

Recurrent pregnancy loss is associated with a pro-senescent decidual response during the peri-implantation window: preliminary rebuttal

We wish to thank the reviewers for the additional comments and we have revised our manuscript further.

Reviewer #2:

Generally, the authors have responded to some of the points raised. However, the critical experiments have not been tackled (see below for details). Therefore, the robustness of the data, its relevance to the situation in vivo and the authors' interpretation of the data, are still questionable. Specifically, the issues focus on three main points:

1. The characterisation of the decidual stromal subpopulations emerging during decidualisation in vitro

According to new Supp. Fig.S4, only cultures from Patient 'a' were sequenced for the entire decidualisation time course. The timepoints during which the different subpopulations begin to emerge, D4, D6 and WD, cells only from patient 'a' are shown. Why have cells from Patients 'b' and 'c' been removed from these crucial timepoints? It is fundamental to have biological replicates as evidence that these populations reproducibly emerge during the in vitro decidualisation experiments, especially if one of the points of this paper is to describe these novel populations. It is not possible to say that this pattern and these transcriptional states are not just specific to patient 'a' or that particular decidualisation experiment?

Authors' response:

Indeed, we generated a detailed temporal map of the transcriptomic changes in a single primary culture and validated the most salient finding, i.e. emergence of two decidual subpopulations, in independent primary cultures. While the responsiveness of primary endometrial stromal cells (i.e. the level of induction of individual genes) may vary from culture to culture, the transcriptomic profiles presented in this study are entirely congruent with previously reported microarray/RNA-seq analysis performed at different time-points (e.g. PMID: 29244071, PMID: 16123151, PMID: 18511503). We further validated the kinetics of the responses by measuring secreted factors in independent cultures. Hence, we are entirely confident that the time-course data presented in our study are representative of responsive primary cultures.

Reviewer #2:

The 'senescent' cells identified have many of the phenotypic hallmarks of differentiated endometrial stromal cells. These include IL-15, ST2, clusterin, CRYAB, HSD11B1, GLRX, GLUL and ALDH1A1. Differentiated stromal cells also stop proliferating and it is not clear why these 'senescent' cells are not just at the terminal end of the differentiation pathway?

Authors' response:

We disagree. Our observations are entirely in keeping with the biology of acute cellular senescence, i.e. that lack of immune cell-clearance of senescent cells leads to propagation of the phenotype through secondary/bystander senescence (exemplified visually by the emergence of islets of SA β G⁺ cells - PMID: 29227245). In a broader context, terminal differentiation and senescence are permanent post-mitotic states in which cells remain viable

and non-proliferative over extensive time frames. Whereas terminally differentiated cells have distinctive phenotypes and highly specialized functions, senescence is a fate (albeit highly variable and heterogeneous) shared by most cell types as a general response to stress. In the endometrium, decidualization is preceded by rapid proliferation (i.e. imparting replicative stress) during the follicular phase, which is arguably unparalleled in any other tissue. It is therefore entirely expected that some cells will not be able to become specialized decidual cells but instead acquire a senescent phenotype.

Reviewer #2:

The *in vivo* validation of the scRNA-seq data for the two decidual stromal subpopulations is performed by ISH for DIO2 and SCARA5 in new Fig. 5d. This figure is unclear. Where are these stromal populations with respect to their localisation within the endometrium? Are they all showing corresponding areas of the endometrium and is the staining pattern for each patient consistent across the entire tissue? The interpretation of these results should be assisted by staining for additional canonical senescent markers such as CDKN2A, on a serial section from the same tissue sample.

Authors' response:

All biopsies were obtained with Wallach Endocell™ endometrial pipelle and thus only superficial endometrium was sampled. The images shown are indeed representative of the biopsies. In the SCARA5^{AVERAGE} / DIO2^{AVERAGE} samples, DIO2⁺ cells appeared to be enriched in the stroma underlying the luminal epithelium as shown in Fig. 5d (middle panel).

Reviewer #2:

2. Comparison of stromal populations in women with RPL. The differences between biopsies from women with RPL and controls is of potential interest. It would then be expected that if the response of the stromal cells to progesterone is defective then IL-15 will also be reduced resulting in fewer NK cells. This may just reflect an overall delayed or altered response to progesterone. How does this data fit with other reports (some from this group) claiming uNK numbers are increased in women with RPL (eg. Quenby et al., Fertil Steril 2005; Tuckerman et al., J Reprod Immunol 2010)?

Authors' response:

Historically, the *a priori* assumption in reproductive medicine was that accumulation of uNK cells in the endometrium signalled an impending 'immune attack' on an implanting embryo. Our previous study (PMID: 29227245) was the first to implicate uNK cells in endometrial homeostasis, both within a cycle and from cycle-to-cycle. Further, none of the earlier studies normalized the abundance of uNK cells for the precise day of cycle.

Confidential information redacted

Reviewer #2:

3. Selective killing of senescent stromal cells by uNK cells

There may indeed be a higher proportion of 'stressed' stromal cells in some women with RPL but to extend this as abnormal chronic senescent cells requiring elimination by uNK cells is an unlikely scenario. uNK cells do not normally kill 'self' cells. The paper that the authors' cite in the rebuttal are their own (Brighton et al., eLIFE 2017). In this manuscript, as in Brighton et al., the protocol used for the isolation of uNK cells isolation is problematic. Their protocol will alter uNK phenotype and killing capacity and in vitro experiments will therefore not reflect the situation in vivo. Purity, viability and activation status of the uNK cells must be assessed before their functional experiments are performed to ensure that the results from their co-culture experiments are not due to in vitro artefacts. This must be done by flow cytometry as these characteristics cannot be properly assessed by CD56 on cytopins, shown in their rebuttal (this data is also not in their previous publication).

Authors' response:

First, the Reviewer appears to have misinterpreted our findings. Our data indicate that uNK cells eliminate acute senescence cells at the time of implantation and thereby prevent chronic senescence. Failure to do so is associated with an increased risk of miscarriage.

Second, our protocol is based on the methods paper of Male et al. (2010) Natural Killer Cells in Human Pregnancy. In: Campbell K. (eds) Natural Killer Cell Protocols. Methods in Molecular Biology (Methods and Protocols), vol 612. Humana Press. It states, "*Some investigators may be concerned that the positive selection procedure may alter the decidual NK cell phenotype. However, CD56 is of unknown function and cross-linking this receptor has never been shown to have an effect on known functions of NK cells. Indeed, anti-CD56 is regularly used as a negative control antibody in NK cell functional assays. MicroBead cocktails are also available for the isolation of dNK cells by negative selection, although these are more costly than the CD56 MicroBeads.*" Several publications have used MACS to isolate uNK cells (PMC5114884 and PMC2709965).

Third, when optimising the uNK cell isolation, we performed RNA-seq on 3 endometrial biopsies in parallel with purified endometrial stromal cells, epithelial cells, and uNK cells. The data are as yet unpublished but the relative expression levels of key uNK cell marker genes are shown in the figure below.

Reviewer #2:

In the author’s rebuttal, they cite several papers to support their claim uNK cells can kill ‘senescent’ cells. However some of these papers (PMID: 24800169, PMID: 17251933) are in the context of another tissue and a pathological situation of tumours, in which it is known that NK cells do kill.

Authors’ response:

NK cell mediated killing of senescent cells is indeed well studied in the context of cancer biology but there is ample of experimental evidence implicating NK cells in non-malignant conditions (e.g. PMC3073300, PMC4118230, PMC3630483, PMC4789586).

Reviewer #2:

Their experimental set-up for Fig. 3d was used in their previous paper (Brighton et al., eLIFE 2017, Fig. 5) and it does not show whether only the senescent cells are being killed. All this assay shows is that there is less SA-β-gal produced and there could be several explanations for this. To demonstrate that uNK cells are able to kill ‘self’ cells that are ‘senescent’, they need to separate the ‘senescent’ cells from the other stromal cells and perform conventional NK killing assays (after assessing uNK status as requested above).

Authors’ response:

As stated before, selective killing of senescent decidual cells by uNK cells was shown using several complementary approaches, including time-lapse microscopy, use of various inhibitors and blocking antibodies, loss of clusterin secretion, and elimination of SA-β-gal activity upon decidualization. To the best of our knowledge, there are no practical or validated techniques to isolate senescent cells in culture (PMID: 27212009). As reported previously, senolytic drugs, such as dasatinib, can be used to deplete senescent cells in culture.

Reviewer #2:

If it proposed that the NK receptor mediating this is NKG2D which binds to stress ligands on stromal cells, this needs demonstrating. ULBPs and MICA/B do not appear in their gene list in Fig 1g. Furthermore, NKG2D is not expressed on all uNK cells (Marlin et al., PLOS One 2012).

Authors’ response:

First, we have an ongoing project on identifying the stress ligands on senescent decidual cells. This is not a trivial exercise as stress ligands are actively cleaved by metalloproteinases. While we have candidate stress ligands, further validation experiments are needed, which are beyond the scope of the current paper. Second, we demonstrate in the current study that cycling endometrium harbours different subpopulations of uNK cells. It appears indeed likely

that different uNK cell subsets may differ in their ability to eliminate senescent decidual cells. Expression of NKG2D, encoded by *KLRK1*, in isolated uNK cells is shown in the above figure.

Additional comments:

Reviewer #2:

The rationale of Fig 4a is not clear in relation to the rest of the paper. The in vivo scRNA-seq profiling of the endometrium still does not show all the immune populations known to be present in the endometrium. NK cells and macrophages are present in the functionalis layer as well as T cells. But there are no B cells in the functionalis. Why do they only find B cells and no T cells in their samples and why are there monocytes? These are only present in blood.

Authors' response:

We sequenced a total of 352 immune cells, which is a relatively low number of cells. We are confident that T cells will be identified if we sequence more samples and, contrary to the claim made by the Reviewer, CD19-positive B-cells can be present in the superficial endometrium. The IC2 population, designated monocytes, encompassed only 16 cells (<5% of total immune cells). Considering the endometrium is a perfused tissue, this finding does not appear unexpected.

Reviewer #2:

Please clarify the following:

Introduction p3L62 – ‘hyperinflammation due to placentation, under physiological conditions.’ What does ‘hyper’ mean?

Authors' response:

In this context, ‘hyper’ means ‘intense’.

Reviewer #2:

Results p7L163 – ‘decidualization is a multistep process that starts with acute auto-inflammatory response.’ The use of the term ‘inflammatory’ is used in this manuscript in relation to placentation and in the initial phases of the decidualisation process. It is not clear why this term is used as inflammation cannot be simply defined by the presence of pleiotropic cytokines such as IL-6. Inflammation is defined by an influx of inflammatory cells (neutrophils, macrophages and other immune cells), deposition of fibrin and edema followed by tissue repair. These are not seen in the early secretory phase (although edema alone is seen in the mid-secretory phase) or the early decidua. The influx of neutrophils occurs at menstruation.

Authors' response:

Decidualization starts indeed during the midluteal phase, is associated with oedema, influx/proliferation of uNK cells and, to a lesser extent, macrophages. The similarities between endometrial remodelling during the implantation window and tissue repair are striking, including the fact that both processes involve a critical role for acute senescent cells. Hence, the term ‘auto-inflammatory response’ appears appropriate.

Reviewer #4

Having read the manuscript, reviewer 2 comments and the authors' rebuttal, I believe that the manuscript by Lucas et al. provides new insights regarding the molecular pathways of human decidualization. The single cell RNAseq analysis performed on in-vitro cultured decidualizing cells is very informative. The identification of specific markers of divergent decidualization response and findings that these are differentially expressed in endometrial biopsies of RPL patients compared to controls are important. However, Reviewer 2 has raised a couple of important points with regards to the main claims of the paper and data interpretation that I tend to agree with as detailed below.

With regards to point 3 raised by the reviewer, I am in agreement with the reviewer that it is unlikely that NK cells eliminate "senescent" decidual cells in-vivo. NK cells do not have "killing" properties against decidual cells and are actually thought to be important in mediating immune tolerance to the fetus. The evidence presented to that effect of eliminating senescent cells by the authors is based on in-vitro studies in both the current work as well as their related referenced paper (Brighton et al., eLIFE). On both occasions, NK cells are shown to be cytotoxic towards decidual cells compared to non-decidualized cells in general. To support their claim, and as suggested by reviewer 2, it would be essential to separate senescent cells from normal decidual cells and show a selective cytotoxic effect against the senescent decidual cells.

Authors' response:

We thank the Reviewer for the positive comments. We do not claim that NK cells "kill" decidual cells nor do we contest the role of these cells in maternal immune tolerance in pregnancy. Our observations do indicate that uNK cells target and eliminate acute senescent decidual cells. As stated in our response to Reviewer #2, there are to our knowledge no practical or validated techniques to isolate senescent cells in culture (PMID: 27212009). However, drugs such as dasatinib should selective target senescent cells and thus recapitulate the effect of co-culturing uNK cells on the secretion of decidual cell and senescent decidual cell markers. We have now tested this hypothesis and decidualized 6 independent primary cultures in the presence or absence of dasatinib for 8 days. As shown in the new Supplementary Figure S7, decidualization of primary EnSC in the presence of dasatinib recapitulated the effects of uNK cell co-cultures, characterized by a marked reduction in SA- β -gal activity and clusterin (CLU) secretion. By contrast, dasatinib had little effect on secretion of sST2 levels, encoded by *IL1RL1*.

Reviewer #4

Moreover, in-vivo evidence to support such claim is seriously lacking, and currently available in-vivo evidence in mice is not supportive of their claim. A lymphoid mice (lacking NK cells) and IL-15 KO mice that also lack decidual NK cells have alteration in spiral artery remodeling but otherwise have normal fertility, decidualization, normal pregnancies and litter sizes (PMID 22187674, PMID 24000237). Thus, the argument that NK cells are required for elimination of senescent cells for pregnancy success is very speculative. In-vivo the uterine microenvironment has many other immune and other cell types that mediate immunomodulation which is likely very different than the in-vitro system. What is needed is to show in-vivo evidence. For example, electron microscopy showing the proximity between NK cells and the senescent cells in human endometrial tissue; A mouse model showing the effects of NK cell enhancement or inhibition with NK cell antibody and showing whether this leads to

elimination or accumulation of the decidual senescent cell phenotype, respectively, would help strengthen such claims.

Authors' response:

The evolutionarily conserved trigger for decidualization is a stress/inflammatory response. In contrast to most mammals with an invading placenta (including mice), this signal is generated endogenously in human endometrium in each cycle, following conspicuous proliferation of stromal cells during the follicular phase. The situation in mice is fundamentally different as the decidualization signal is exogenous – i.e. triggered by embryo implantation (recapitulated by local trauma of the endometrium or exposure to oil etc.). Hence, we do not anticipate that our finding of acute decidual senescence in cycling human endometrium is conserved in mice. However, we would like to point out that the decidua in alymphoid mice is not normal, as stated by the Reviewer, but displays hypocellularity and sometimes necrosis (PMID: 10899912), an observation entirely compatible with a role for uNK cells in decidual homeostasis. As for the *in vivo* situation, we agree that the current lines of evidence are indirect. For example, analysis of paired endometrial biopsies obtained in different cycles revealed a strong inverse correlation between the abundance of uNK cells and the expression of senescent decidual marker genes, such as *DIO2* (unpublished data). We appreciate the suggestion of this Reviewer to use imaging and we will explore this approach in future studies.

Reviewer #4

With regards to the reviewer's additional comment on the immune populations in the scRNAseq analysis, the presence of a small subset of B-cells and some monocytes is likely due to some contamination from circulating leukocytes, as eluded to by the authors' answer in the rebuttal. As the authors rightfully acknowledge, the small number of immune cells analyzed (352 cells only) limits the depth of the analysis. Overall, the large abundance of NK cells is expected. In Figure 4, the population identified as 'monocytes' (IC2) actually has strong expression of CD4, a T cell marker, as well as expression of several dendritic cell markers. This suggests to me that IC2 population may be misidentified and likely contains the T cells that the reviewer found to be missing. This can probably be addressed by re-analysis of that cluster to look for typical T cell markers.

Authors' response:

The new Supplementary Table 7 now lists the marker genes for each immune cell cluster, the average gene expression of each immune cell cluster, and the CIBERSORT analysis. The latter analysis identified IC2 as monocytes. Although IC2 cells do express higher *CD4* levels compared to the other endometrial immune cell populations, they are devoid of other pan T cell markers, including *CD2* (no detectable transcripts), *CD3D* (no detectable transcripts), *CD5* (no detectable transcripts) and *CD7* (no detectable transcripts). We now indicate in the text that identification of IC2 as monocytes was based on CIBERSORT analysis. We hope that the inclusion of the new Supplementary Table 7 now resolves this issue.

Reviewer #4

With regards to point 1 of the reviewer, I agree with the reviewer that *in vivo* validation of the scRNA-seq data would be greatly enhanced by addition of canonical senescent markers such as *CDKN2A*, to the current validation which is based on ISH for *DIO2* and *SCARA5*.

Authors' response:

We are planning follow-up scRNA-seq analysis to characterize the endometrial state further based on the relative expression of *DIO2* and *SCARA5*. In culture, the expression of *CDKN2A* (coding p16INK4) and *CDKN2B* (p15INK4) correlate with *DIO2* expression. In *vivo*, the temporal expression of *DIO2*, characterised by peak expression during the late-luteal phase, closely mirrors that of other canonical senescent markers, including p53, SA β GAL, p16INK4, and *CDKN2B* (GEO ID: 24491063)

Reviewer #4

With regards to point 2 of the reviewer, the literature regarding NK cells abundance in the endometrium of women with RPL is conflicting, and the authors' response is satisfactory in my mind.

Authors' response: Thank you.

Reviewer #5

Point 1 (Supp FigS4). The reviewer asks about data from other patients to make sure data are reproducible. Author says data are "representative of responsive primary cultures". I would agree with the Reviewer here that results from other patients would make the data stronger.

Authors' response:

Indeed, we generated a detailed temporal map of the transcriptomic changes in a single primary culture and validated the most salient finding, i.e. emergence of two decidual subpopulations, in independent primary cultures. While the responsiveness of primary endometrial stromal cells (i.e. the level of induction of individual genes) may vary from culture to culture, the transcriptomic profiles presented in this study are entirely congruent with previously reported microarray/RNA-seq analysis performed at different time-points (e.g. PMID: 29244071, PMID: 16123151, PMID: 18511503). We further validated the kinetics of the responses by measuring secreted factors in independent cultures. Hence, we are entirely confident that the time-course data presented in our study are representative of responsive primary cultures.

Reviewer #5

Point 2. About senescent vs terminally differentiated cells. I think the Author's response here is reasonable.

Authors' response: Thank you.

Reviewer #5

Point 3. *in vivo* validation of the scRNA-seq data. The Author's response seems adequate here.

Authors' response: Thank you.

Reviewer #5

Point 4. Comparison of stromal populations in women with RPL. This response seems reasonable too and the associated data (for information only) are convincing.

Authors' response: Thank you

Reviewer #5

Point 5. Selective killing of senescent stromal cells by uNK cells. I think the Author's response about cell preparation is fine, however the Reviewer's request to show FACS data of uNK cells is also reasonable and ought to be included.

Authors' response:

As requested, we performed FACS analysis on uNK cells isolated first by MACS from the supernatant of 4 independent freshly established primary EnSC cultures. The results are presented in the new Supplementary Figure 6. Approximately 86% of cells in the positive fraction were confirmed to be viable uNK cells.

Reviewer #5

Point 6. Claim that uNK cells can kill 'senescent' cells. I'm afraid I know of no published data showing this directly. The papers mentioned by the Author are indeed about non-malignant conditions, however not from the uterus, so I am not sure they really help the point of the Author, since every tissue-resident cell will have a tissue-specific biology.

Authors' response:

We are not sure what 'direct' evidence we can provide in cycling human endometrium but we appreciate the suggestion to pursue an imaging approach. As aforementioned, analysis of paired endometrial biopsies obtained in different cycles revealed a strong inverse correlation between the abundance of uNK cells and *DIO2* expression (unpublished data).

Reviewer #5

Point 7. Their experimental set-up for Fig. 3d was used in their previous paper (Brighton et al., eLIFE 2017, Fig. 5) and it does not show whether only the senescent cells are being killed. The Author's response seems reasonable here.

Authors' response: Thank you

Reviewer #5

Point 8. NKG2D and its ligands. The Reviewer is right here that there is no data supporting the involvement of this receptor-ligand and the authors may want to show these data or not mention this R-L interaction at all, or make sure it is only a speculation.

Authors' response:

As aforementioned, the nature of the stress ligands express by senescent decidual cells and the mechanisms of uNK cell recognition is under further investigation. Also, we have now compared the effects of uNK cell co-culture with the effects of dasatinib (new Supplementary Figure S6) on SA β G activity, and clusterin and sST2 secretion. Both approaches yielded comparable results, which should instil further confidence that uNK cell selectively target senescent decidual cells.

Reviewer #5

In conclusion, and having read the interesting paper, my suggestion is to accept it but perhaps change the title (and the relevant text in the paper) to a more cautious:

“Recurrent pregnancy loss may be associated with a pro-senescent decidual response during the peri-implantation window”. Rather than is associated with...

Also, I have noticed discrepancies between the Vento-Tormo paper (Nature 2018) and the RNAseq data published here from the three subsets of dNK cells. Specifically, in the Vento-Tormo paper EPAS1, GZMA and GNLY are highly expressed in dNK1, whereas in this paper they are highly expressed in dNK2. Perhaps the Authors ought to comment on this, rather than misleadingly pointing at the similarities only.

Authors' response: We believe that our observations are robust. Further, the title of the manuscript does not claim or infer causality, merely an association. We identified and labelled the 3 uNK subsets in cycling endometrium before the publication of the Vento-Tormo paper. The term 'match' on line 248 was unfortunately as it could be read that NK1-3 match dNK1-3 identified by Vento-Tormo and colleagues. We have rephrased this statement to avoid confusion.

REVIEWERS' COMMENTS:

Reviewer #4 (Remarks to the Author):

Supplementary Figure 6: Data shown here are convincing of uNK cell purity. However data are from one of 4 samples. Authors should state, at least in the figure legend, what the % of uNK cells was in the other 3 samples used for experiments.